# The 3D Printing of Nanocomposites for Wearable Biosensors: Recent Advances, Challenges, and Prospects

**DOI:** 10.3390/bioengineering11010032

**Published:** 2023-12-27

**Authors:** Santosh Kumar Parupelli, Salil Desai

**Affiliations:** 1Department of Industrial and Systems Engineering, North Carolina Agricultural and Technical State University, Greensboro, NC 27411, USA; sparupel@ncat.edu; 2Center of Excellence in Product Design and Advanced Manufacturing, North Carolina Agricultural and Technical State University, Greensboro, NC 27411, USA

**Keywords:** 3D printing, biomedical, health monitoring, nanocomposites, wearable biosensors

## Abstract

Notably, 3D-printed flexible and wearable biosensors have immense potential to interact with the human body noninvasively for the real-time and continuous health monitoring of physiological parameters. This paper comprehensively reviews the progress in 3D-printed wearable biosensors. The review also explores the incorporation of nanocomposites in 3D printing for biosensors. A detailed analysis of various 3D printing processes for fabricating wearable biosensors is reported. Besides this, recent advances in various 3D-printed wearable biosensors platforms such as sweat sensors, glucose sensors, electrocardiography sensors, electroencephalography sensors, tactile sensors, wearable oximeters, tattoo sensors, and respiratory sensors are discussed. Furthermore, the challenges and prospects associated with 3D-printed wearable biosensors are presented. This review is an invaluable resource for engineers, researchers, and healthcare clinicians, providing insights into the advancements and capabilities of 3D printing in the wearable biosensor domain.

## 1. Introduction

With the advancement of technology, there has been a meteoric development of wearable sensors in telehealthcare and biomedical domains. Wearable sensors have applications across various areas such as healthcare, biomedical, industrial automation, caregiving, sports, military, smart clothing, crisis management, and health monitoring. The significant development of sensing materials, flexible electronics, the Internet of Things (IoT), and cloud architectures has led to a profound augmentation of these devices for healthcare services. Different types of wearable sensors were developed for monitoring mechanical, biological, temperature, chemical, humidity, etc., signals. The compound annual growth rate (CAGR) of wearable sensors is projected to grow by 18.3%, from USD 840 million in 2021 to USD 3.7 billion in 2030 [1]. Wearable biosensors (WBSs) are compact electronic widgets that coalesce sensors into/or with the human body in clothing, gloves, implants, and tattoos, performing in vivo sensing, data collection, and analysis with human-interface portable devices [2,3,4,5,6,7,8,9,10]. The crucial biomarker elements of the human body, which include heartbeat, blood pressure, brain waves, body temperature, oxygen saturation, glucose level, and breathing patterns, can be recorded and monitored with wearable biosensors. Tracking these essential elements has the capability to enhance diagnosis, prognosis, postoperative treatment, and the treatments of chronic conditions for human well-being [11,12,13,14,15]. WBSs help healthcare practitioners and providers provide a top-notch user experience with customized patient-specific interventions while minimizing the clinical visit frequency [3,16,17,18,19,20]. Leland C. Clark, who is known as the “father of biosensors”, invented the biosensor device named the Clark electrode back in 1956 [21]. This device estimates oxygen in blood, water, and other liquids with applications in biomedical, environmental, and industrial areas. In 1969, Guilbault and Montalvo developed an inaugural potentiometric biosensor specifically designed for urea detection, a significant improvement in the medical field. This later development was a vital moment in biosensor technology, which followed Leland Clark’s pioneering work on amperometric enzyme electrodes in 1962. Based on the above-mentioned foundational discoveries, the first commercial biosensor was developed by Yellow Spring Instruments (YSI) in 1975, marking a historic milestone in the practical applications of biosensor technology [22,23]. Table 1 represents some of the commercially available biosensors in the market.

GlucoWatch^®^ G2 Biographer was the earliest commercialized wearable glucose sensor approved in 1999 by the U.S. Food and Drug Administration [25]. The commercial adoption of the biosensors market has been relatively slow due to various key factors such as high cost, manufacturing technologies, materials, stability, and quality control. Ongoing research and development endeavors play a vital role in ensuring the continuous advancement of biosensor technology [26,27]. As time advances, these efforts will enable the development of accurate, more precise biosensors adapted for specialized high-end applications. Several of the key players currently working on the development of biosensors include Abbott Point of Care Inc. (Kansas City, MO, USA), Siemens Healthcare Diagnostics Inc. (Los Angeles, CA, USA), Medtronic Diabetes (Devonshire Street Northridge, CA, USA), Applied Enzyme Technologies (Maharashtra, India), Biozyme Laboratories (St. Joseph, MO, USA), Nova Biomedical, LifeSensors Inc. (Mississauga, ON, Canada), Affinity Sensors (Santa Barbara, CA, USA), VeriChip (Delray Beach, FL, USA), BIAcore AB (Uppsala, Sweden), GeneOhm Sciences (Baldwin, MD, USA), and Sensor Tech. Ltd (Collingwood, ON Canada). The evolution timeline of the biosensing technologies leading to modern wearable biosensors is illustrated in Figure 1 [28].

Biosensors are devices that combine biological elements and physicochemical components to generate a specific measured signal by detecting an analyte in the process. Figure 2 illustrates the schematic interpretation of biosensor components [2,29]. A standard biosensor comprises two basic functional units, namely a bioreceptor or biorecognition element (enzyme, DNA, peptide, antibody, nucleic acid, etc.) and a physicochemical transducer, for example, an electrochemical, thermal, optical, or piezoelectric device. The bioreceptor is used to selectively identify the target analyte, whereas the transducer is used for the translation of a biorecognition outcome into a detectible signal [22,29,30]. These devices were primarily intended and developed for in vitro or single-use home measurements such as GlucoWatches, point-of-care instruments, and glucometers [2]. As shown in Figure 1, past advancements have led to significant progress in the development of modern wearable biosensors for noninvasive biomonitoring in a wide range of biomedical and healthcare applications [31]. Biosensors are suitable for wearable applications because of their low/minimal cost, high specificity, and nominal power requirements. Biosensors have applications in drug discovery, pollutant detection, disease monitoring, and disease-causing microorganisms, and they are also used to identify bodily fluid markers such as blood, urine, saliva, and sweat in a variety of head-to-toe application sites [32,33,34,35,36,37].

As mentioned above, wearable biosensors have the potential to play a vital role in research studies as well as human life. Wearable biosensors are employed for detecting human physiological parameters such as glucose, blood pressure, respiratory parameters, tactile parameters, oxygen saturation, heart rate, respiration rate, body motion, skin temperature, and brain activity [38,39]. These devices can be manufactured with numerous manufacturing methods such as injection molding followed by soft lithography, laser ablation, lamination, semiconductor lithography followed by electroplating or coating, and planar printing. However, there are certain limitations on how these technologies can be used for the fabrication of flexible, complex structures and wearable biosensors. These include challenges such as manufacturing scalability deficiency, higher production costs, poor robustness, and restricted extensibility. Lately, additive manufacturing (AM), commonly known as 3D printing, has steadily been developed and has managed to overcome the above-mentioned drawbacks of traditional manufacturing techniques. AM is a highly adaptable manufacturing technique that is used to fabricate 3D structures directly from computer-aided design model files in a layer-by-layer fashion, with minimal material wastage and a high degree of freedom. AM has accelerated significant transformations across many domains, such as macro/microarchitectures, biomedical devices, wearable biosensors, flexible sensors, integrated devices, patient-specific treatment, and microfluidics [40,41,42,43,44,45,46,47,48,49,50,51,52,53,54]. The fusion of AM with wearable biosensors enables groundbreaking development in the realm of customized biomedicine, healthcare diagnostics, and treatment [55,56,57,58,59,60,61,62,63,64,65,66,67]. The integration of these two arenas offers unique prospects to develop multipurpose, personalized, and flexible biosensor platforms that can impeccably integrate with the human body for the real-time monitoring of physiological parameters. In the past few decades, the advancement and numerous research contributions from various scientists and industries have led to several momentous enhancements in AM systems, including the fabrication of novel materials such as nanocomposites, higher resolution, and improved printing speeds. Moreover, among the prospective applications of AM that have been developed in recent years, the fabrication of wearable biosensors has garnered substantial attention [68,69]. AM processes are categorized based on printing technology and the working materials. Different classifications of the AM process include fused depositing modeling, direct-write printing, inkjet printing, binder jetting, powder bed fusion, selective laser sintering, directed energy deposition, lamination, stereolithography, digital light processing, two-photon polymerization, and continuous liquid interface process. There are still many untapped applications of 3D printing yet to be explored [70]. Due to some of the significant limitations of 3D printing technologies such as strength competence and functionality, the commercialization of 3D-printed objects is not feasible yet. Thus, one of the propitious approaches is to incorporate nanoparticles or fillers into the polymer to reinforce a 3D-printed object [50,71,72,73,74]. Lately, extensive research has been conducted to develop more functional structures by incorporating nanoparticle materials [43,69,75,76,77,78,79,80,81,82,83].

This review presents the significant developments of 3D printing technologies in the past decade for fabricating novel next-generation wearable biosensor structures using nanocomposite materials. It also provides research progress in wearable biosensors for various applications such as temperature sensing, continuous glucose and blood monitoring, brain activity analysis, strain analysis, and biomarker detection. A comprehensive review of 3D-printed wearable biosensors such as electrocardiograms (ECGs); electroencephalograms (EEGs); and blood pressure, glucose, oxygen saturation (SpO2), sweat, textile, and respiratory biosensors is reported. This review also covers general wearable biosensor deployment challenges in terms of factors such as accuracy, user acceptance, and data security. The focus of this paper is to provide an overview of recent developments and contribute to the understanding of the technical foundations of 3D-printed wearable biosensors and their inferences for facilitating advanced healthcare and enhancing human well-being.

## 2. Nanocomposites

Nanocomposites are composites that consist of at least one phase within the nanometer range (10^−9^ m). Nanocomposites are materials comprising nanoparticles that are incorporated into a standard matrix material. The nanomaterials used in nanocomposites include nanoparticles, nanofibers, and nanolayered silicates [84,85,86]. The addition of nanoscale particles drastically improves material properties such as electrical and mechanical properties, thermal conductivity, toughness, and strength compared to bulk components [87]. Nanocomposites, which are composed of nanoparticles and matrix material, have gained substantial attention due to their unique characteristics and potential applications [68,77,88,89]. Depending on the matrix material used, nanocomposites can be categorized into three types, as shown in Table 2. These include polymer matrix nanocomposites (PMNCs), ceramic matrix nanocomposites (CMNCs), and metal matrix nanocomposites (MMNCs). Table 2 provides detailed examples of different types of nanocomposite materials belonging to these three classes, along with their respective properties.

### 2.1. Polymer Matrix Nanocomposites (PMNCs)

Nanocomponents in PMMC are typically fillers termed nanofillers and classified as 1D—linear (e.g., carbon nanotubes), 2D—layered (e.g., montmorillonite), and 3D—powder (e.g., silver nanoparticles) [91]. PMNCs can be synthesized with techniques such as the intercalation of the polymer or pre-polymer from solution, melt intercalation, in situ intercalative polymerization, and template synthesis (sol–gel technology). The characteristics of PMNCs include improved mechanical properties (high abrasion resistance), lower gas permeability, and high thermal stability. Enhancements in the properties of polymer nanocomposite materials have led to the development of abundant industrial applications, for example, fuel cells, fuel tanks, plastic containers, packaging, power tool housing, and solar cells [88,90].

### 2.2. Ceramic Matrix Nanocomposites (MMNCs)

Ceramic matrix nanocomposites are materials in which one or more discrete ceramic phases are added to improve wear resistance and thermal and chemical stability. In CMNC production, the ceramic matrix is reinforced with energy-dissipating elements such as fibers, particles, or platelets to enhance fracture durability and reduce brittleness [86,92]. CMNCs can be synthesized with techniques such as the melt spinning of hybrid precursors followed by the pyrolyzation and curing of the fibers, powder processing, methods using polymer precursors, vapor techniques (CVD and PVD), chemical methods (colloidal, precipitation, and sol–gel methods), and spray pyrolysis. The advantageous properties of nanocomposites such as improved strength, creep resistance, toughness, stability, and hardness have enabled the development of advanced prosthetics, medical devices, and implants with the amalgamation of bioactive properties with mechanical properties. Some of the popular applications of MMNCs are microelectronic and microelectromechanical systems, chemical sensors, and biomedical devices [90,93].

### 2.3. Metal Matrix Nanocomposites (MMNCs)

Metal matrix nanocomposites (MMNCs) are multiphase materials consisting of a flexible metal or alloy matrix in which certain nanosized fortification material is implanted. The properties of MMNCs include high strength, ductility, modulus, and toughness. In MMNC production, the metal materials used include Al, Mg, Pb, Sn, W, and Fe, but the reinforcements are the same as for CMNCs and PMNCs [86,92]. MMNCs can be synthesized with techniques such as vapor techniques (CVD and PVD), chemical methods (colloidal and sol–gel techniques), rapid solidification, electrodeposition, liquid metal infiltration, and spray pyrolysis. Some of the popular applications of MMNCs include electronic packaging, structural materials, wear-resistant materials, thermally graded coatings, catalysts, and optical/magnetic storage materials [87,90].

Nanocomposite materials demonstrate enhanced properties due to an exceptionally high aspect ratio. The integration of nanoparticle reinforcements enables improved performance as compared to monolithic or microcomposites. Nanomaterials, which include nanorods, nanofibers, nanowires, and nanotubes, are more promising for the different types of matrix nanocomposite materials for the 3D printing of wearable biosensors [94]. These nanomaterials have a high surface-to-volume ratio, which makes them an ideal candidate for sensing applications [87,90]. Based on the desired material properties, the relevant nanocomposite materials (PMNCs, CMNCs, and MMNCs) can be utilized for various applications such as biomedical devices, healthcare applications, solar cells, aerospace applications, supercapacitors, etc. MMNC materials improve material properties such as ductility and toughness. Therefore, MMNCs are more suitable for applications with high strength in shear/compression and extreme temperature resistance. The reinforcement of nanofibers in ceramic matrices enables the advancement of CMNC materials for applications with superior failure properties and high toughness. In contrast, polymer matrix nanocomposites (PMNCs) possess enhanced electrical conductivity and colloidal stability. PMMNC materials improve the biodegradability of materials, which is essential for biomedical and healthcare applications [86]. Nanomaterials are capable of both improving the detection of moieties and accelerating signal transduction. Moreover, the slightest change in response can be captured effectively. The major objective of incorporating nanomaterials is to elevate biosensor accuracy and efficiency by augmenting sensor responses, considerably lowering the limit of detection (LOD), and reducing processing times. Due to the unique capabilities of nanomaterials such as their distinctive size and shape-dependent attributes, and particularly due to their role in energy band-gap modulation, these materials provide rapid and precise analyte detection. For example, traditional glucose detection techniques can be significantly enhanced by functionalizing enzymes on graphene sheets or carbon nanotubes, which in turn increases the detection sensitivity and reduces the LOD. Furthermore, the incorporation of nanomaterial-based alternatives such as polymer/metal/ceramic nanoparticles, quantum dots, nanowires, and nanorods could result in improved performance of wearable biosensor devices [95]. These remarkable abilities owing to nanomaterial interactions with analytes enable effective biosensing, especially significant for biomedicine and healthcare domains. The utilization of nanomaterials in biosensing technology is considered a cutting-edge advancement in the detection capabilities of wearable biosensors [92].

## 3. Additive Manufacturing (3D Printing) Processes

Additive manufacturing (AM), widely known as 3D printing, is used to build 3D structures in a layer-by-layer fashion using different materials such as polymers, metallic pastes, composites, and ceramic based on a 3D computer-aided digital data file [96,97]. The overview of various AM processes is illustrated in Table 3. AM has been given numerous terms such as additive fabrication, 3D printing, layered manufacturing, additive techniques, free-form fabrication, digital manufacturing, additive processes, and additive layered manufacturing [98]. The ASTM standard definition of AM is the “process of joining materials to make objects from 3D model data usually layer-by-layer, as opposed to subtractive manufacturing technologies such as traditional manufacturing”. There are seven categories of additive manufacturing processes, namely photo-polymerization, extrusion-based systems, powder bed fusion, material jetting, binder jetting, directed energy deposition, and sheet lamination processes, which are shown in Figure 3. According to ASTM F2792, AM processes are categorized into seven types as mentioned above [99,100]. Fused deposition modeling techniques have applications in hearing aids, surgical instruments or tools, dental implants, and pharmaceutical applications. Stereolithography techniques have applications in prosthetics, implantable devices, orthodontics, surgical planning, drug delivery, and dentistry. Selective laser sintering is used in tissue engineering, as well as for the fabrication of orthopedic devices and surgical instruments or tools. Binder jetting is used for manufacturing surgical instruments or tools, medical models, and drug implant prosthetics. Electron beam melting is used to manufacture hip and knee replacements, cranial implants, dental implants, and surgical instruments or tools. Digital light processing has applications in tissue engineering, surgical planning, and medical imaging. The most popularly used 3D printing techniques for wearable biosensors are fused deposition modeling, selective laser sintering, inkjet 3D printing, stereolithography, two-photon polymerization, and laminated object manufacturing [101]. A detailed overview of different types of 3D printing processes with technical details is presented in Table 3.

Vat photopolymerization (VP): This method is used to build 3D structures based on liquid resin, which solidifies when exposed to ultraviolet light. UV light is used to cure photosensitive resin to fabricate the required parts based on a 3D CAD file. It has high accuracy and a smooth surface finish. The applications of the VP method include medical instruments, electronics, aerospace applications, injection molds, defense applications, and form-fit objects.

Extrusion-based systems (EBSs): In these systems, adequate pressure is applied to push the raw materials out of the hot extrusion nozzle, and 3D objects are fabricated by depositing the molten material onto the substrate from the bottom up sequentially. The method is relatively slow compared to other AM processes and has restricted layer thickness accuracy. Applications of EBSs are found in the medical field as well as a variety of industries such as architecture, aerospace, automotive, jewelry, and art.

Powder bed fusion (PBF): In the PBF process, either an electron beam or thermal energy is employed as an energy medium to melt and fuse small powder particles to build 3D structures. It does not require any support while building the overhang and unsupported structures because the residual powder provides the required support. The completed 3D parts tend to be rough and porous depending on the material used. PBF applications are found in various industries such as medical, electronics, aerospace, automotive, and military sectors.

Material jetting (MJ): In the MJ process, inkjet printheads are used to dispense raw material onto the build platform to build the final 3D object. Multiple printheads can be utilized to build a 3D object with diverse materials. It allows for the fabrication of 3D parts with high accuracy, fine finishing, and various colors. Applications of the MJ process include medical devices, prototypes for form-and-fit testing, jewelry, and rapid tooling patterns.

Binder jetting (BJ): In this process, two materials, namely powder and liquid materials, are utilized to manufacture 3D objects layer by layer. The liquid material acts as an adhesive to bind the layers together to fabricate a solid 3D object. Thus, 3D parts with various materials such as metals, composites, ceramics, sand, and polymers can be fabricated. The finished 3D objects are fragile with inadequate mechanical properties. Applications of the BJ process include prototypes, consumer goods, casting patterns, and architectural products.

Directed energy deposition (DED): In the DED AM process, the powder is deposited and fused with a laser, electron beam, or plasma arc simultaneously to manufacture a 3D part. Typically, DED is used to repair or add additional elements to the current component. The whole printing process should be carried out in an inert atmosphere or vacuum. Fully dense parts with substantially controllable microstructural features can be fabricated with this process. Applications of this method include aerospace applications, military applications, prototypes, and medical implants.

Sheet lamination process (SLP): In SLP, thin sheets of raw material are bonded together to build the 3D parts. Materials such as plastics, metal, and paper are used in this process. Various mechanisms such as thermal bonding, adhesive bonding, ultrasonic welding, and clamping are utilized to bind the sheets together. It is a fast and cost-effective process. Applications of SLP include automotive devices, prototypes, and aerospace applications.

Rapid developments in 3D printing technologies over the past two decades have paved the path toward the development of numerous biomedical applications, such as wearable biosensors [75,82,104,105,106,107,108,109,110,111,112,113].

Many of the above-mentioned 3D printing technologies have been gradually established due to the significant enhancements in manufacturing processes, feature size, printing speed, material, and resolution with respect to layer fusion. Thus, 3D printing technology for the fabrication of sensors is superior compared to conventional manufacturing processes. Remarkably, advanced multifunctional materials are being investigated by numerous researchers (Figure 4) for application in 3D-printed wearable devices, providing enhanced solutions to physicians, healthcare specialists, and patients. Using nanocomposites in advanced processes such as 3D printing increases the possibilities of applications across various sectors and holds significant promise for the fabrication of multifunctional nanocomposites [75,82,107,108,109,110,111,112,113]. Wearable biosensors such as wearable electronics, artificial skin, pressure sensors, glucose sensors, strain sensors, tattoo sensors, etc., have raised considerable interest in the healthcare arena for real-time health monitoring applications [114,115,116]. The success of these intelligent wearable devices is based on printed architectures achieved with advanced 3D printing processes [55,56,117,118,119]. With this in mind, we review the extremely significant contributions of 3D printing processes toward the advancement of wearable biosensors.

## 4. Recent Progress in Additive Manufacturing (3D Printing) of Wearable Biosensors

Notably, 3D printing technology has emerged as a transformative approach in the field of wearable biosensors, enabling the fabrication of intricate sensor structures directly onto flexible and biocompatible substrates. This review reports the recent advancements in 3D-printed wearable biosensors, highlighting their contributions and applications in healthcare and personalized monitoring. In this section, the research progress of 3D-printed wearable biosensors, which include electrocardiograms (ECGs); electroencephalograms (EEGs); and blood pressure, glucose, oxygen saturation (SpO2), sweat, textile, and respiratory biosensors are presented. The scope of this review article is limited to the above-mentioned wearable biosensors.

### 4.1. Electrocardiogram (ECG) Biosensors

ECGs are used to assess the electric activity and the rhythm of the heart by detecting the electric signals generated by the heart each time it beats. ECGs are used to diagnose and monitor conditions affecting the heart. As shown in Figure 5, recent advances in 3D bioprinting have remarkably enabled the design of adaptable and versatile ECG biosensors as wearable biosensors with high accuracy and comfort for monitoring heart impulses. These sensors can also be integrated into clothing or applied as adhesive patches, which can deliver seamless heart rate and rhythmic data for the prediction and early diagnosis of cardiovascular disorders [120]. Li et al. proposed employing photocuring techniques to develop highly customizable 3D-printed graphene/polymer nanocomposites for smart clothing capable of electrocardiography and electromyography. The findings show that sophisticated 3D-printed resin components can be successfully integrated into smart apparel for monitoring human physiological conditions [121]. To produce precise electrode shapes on a large scale with high-resolution details in a repeatable manner, Salvo et al. presented a 3D printing process involving design, manufacturing, and testing, which is appropriate for quick and affordable output. The results of their study indicate that these dry electrodes are efficient sensors for capturing ECG and EEG data [122]. Another study proposes a novel double-network (DN) gel approach to develop conductive, durable, adhesive, and injectable hydrogels. The developed stretchable and flexible functional hydrogel materials present exceptional features, such as adhesion, strain-sensing performance, and biocompatibility. Moreover, for detecting substantial body movements and monitoring human heartbeats, flexible and elastic hydrogel materials are extremely useful for wearable device applications [123]. An ergonomic and affordable user-friendly Holter electrocardiogram (ECG) was developed by Demircioglu et al. that enables remote patient monitoring. The device can send SMS messages and can store data under specialized conditions. Only under certain specific conditions can data recording be carried out to optimize battery usage, storage, and processing performance. Various types of medical disorders can be identified by including features such as email capabilities, GPRS, and mobile app monitoring [124]. Abdou et al. demonstrated the viability of dry electrodes for newborn single-lead ECGs for routine standard check-ups and remote monitoring applications. The results of their study revealed an accuracy of 92.1% in measuring rapid heart rate and transmitting single-lead ECG data using a 3D-printed ECG monitoring device with dry electrodes. This work represents a substantial advancement in newborn-centric monitoring devices that are available to all users [125].

Alsharif et al. studied the fabrication of wearable and semi-flexible dry electrodes using a copper-based filament (Electtrifi) with high conductivity (0.006 Ω·cm) for biomedical applications like ECG signal monitoring. A fused-filament 3D printing technique was utilized, and the effect of printing process parameters and different surface structures of the dry ECG electrode, namely flat, rough, and concentric, on the electrical performance of the fabricated dry electrode was investigated. The results illustrated that the flat dry electrode structures of Tbed = 80 °C and Tnozzle = 140 and 150 °C had a significant impact on the dry electrode’s conductivity, functionality, and impedance measurements [128]. Foster et al. investigated the design and manufacturing of 3D-printed, fur-friendly conductive electrodes for noninvasive canine ECGs used to monitor heart rate and heart rate variability in animals. Preliminary in vivo results highlighted the feasibility of these electrodes to measure heart rate activity signals in canine puppies. The 3D-printed electrodes showed similar electrochemical properties equivalent to commercial sticky electrodes utilized in veterinary clinics [129]. Ahmmed et al. demonstrated a novel 3D printing technique to create conductive metal surfaces for subcutaneous ECG implants in small animals, thereby enhancing ECG monitoring technologies. The in vivo proof-of-concept experimental results were promising, showing an accuracy equivalent to that of a benchtop and commercial ECG system. The reported medical implant device can be a valuable tool for veterinarians, animal scientists, and biomedical researchers to continuously examine physiological parameters in real-time with a superior resolution [130]. Larrea et al. developed photocurable and conductive polymer inks to fabricate conductive shape-form hydrogels for flexible wearable biosensing applications using digital light processing, which is a 3D printing technique. The 3D-printed hydrogels showed high electrical conductivity (range between 10^−1^ and 10^−2^ S cm^−1^) and thus can be used as bioelectrodes for ECG and EMG recordings with improved detection signals related to commercial Ag/AgCl medical electrodes [126].

### 4.2. Electroencephalogram (EEG) Biosensors

The monitoring of brain activity traditionally involves the placement of metal electrodes on the scalp to capture small electrical potentials generated by neuronal activity within the brain using a noninvasive method called electroencephalogram (EEG) technology [131]. Its distinct advantages include high time resolution, enabling the tracking of brain events with millisecond accuracy, and the potential for portable neuroimaging, extending its utility to real-world scenarios outside clinical and lab settings [120,132,133,134]. In recent years, with progress in 3D bioprinting, this area has seen a revolution in cognitive research and neural diagnostic applications, as illustrated in Figure 6. Owing to EEG biosensor innovations, they are easily adjusted to the contours of the scalp, making them extremely comfortable, and the signal quality and user comfort are greatly enhanced [135].

Xing et al. introduced a flexible dry EEG electrode with low cost that is manufactured via 3D printing, demonstrating its ability to capture steady-state visual evoked potentials (SSVEPs) and proposing approaches for the advanced optimization of conductive materials. This research work provides some guidelines on the conductive material formulation for EEG manufacturers to enhance their mechanical and electrical properties [136]. Cho et al. developed a 3D-printed sensor that serves as a time- and money-efficient alternative to conventional production techniques. This sensor can detect a variety of bioelectric signals (EEG and ECG) and generate similar electrophysiological data. This 3D printing technique has the potential to accelerate the progress of modest noninvasive sensors via affordable equipment and offers physiologists a cost-effective solution compared to traditional microfabrication techniques [141]. Ho et al. presented novel 3D printing techniques and soft materials that enable high resolution, flexibility, and electrical conductivity for patient-specific wearable devices. These 3D-printed sensors can effectively and accurately record both passive and active biosignals (EMG, EDA, EEG, and body strain) [140]. Ramasamy et al. demonstrated the 3D fabrication of a low-cost flexible wearable fractal-based biosensor system for neurocardiology and healthcare applications. The advancement in 3D fabrication methods enables the development of novel wearable biosensors with complex features and superior mechanical properties [42]. Schuhknecht et al. developed a novel EEG sensor for recording and monitoring the human brain’s electrical activity. The fabricated electrodes were flexible and amenable enough to be worn on the head for extended periods of time without exerting strain on the user. The 3D-printed sensing electrodes had comparable signal quality to that of the commercial electrodes while providing a far more enhanced comfort level [142]. Velcescu et al. demonstrated the fabrication of flexible EEG electrodes to record brain activity using the 3D printing method. Furthermore, a silver/silver–chloride coating was used to enhance the sensing performance of the EEG. The new electrodes exhibited enhanced contact impedance and reduced contact noise compared to electrodes developed in their prior work [138]. Krachunov et al. reported a novel low-cost 3D printing method for the design and manufacture of dry electrodes to record brain activity noninvasively. This enables rapid and cost-effective electrode manufacturing and unveils the possibility of producing customized patient-specific electrodes. The performance of the developed 3D-printed electrodes is appropriate for brain–computer interface applications when compared to commercial wet electrodes [137].

### 4.3. Blood Pressure Biosensors

Bioprinted blood pressure biosensors allow for the continuous monitoring of blood pressure levels noninvasively. Real-time data can be collected for hypertension management by integrating biosensors into wearable patches or cuffs. As per North American, Japanese, Chinese, and European hypertension guidelines, out-of-office BP monitoring has been recommended as an ideal technique in the past 30 years [143,144]. Recent developments in wearable health solutions, specifically blood pressure biosensors, as shown in Figure 7, are currently revolutionizing the healthcare landscape. With the emergence of further granular and longitudinal health data, it has become conceivable to track and analyze every small change in individual health conditions [145,146,147,148]. Li et al. used an electric field-assisted 3D printing technology to fabricate in situ-poled ferroelectric artificial arteries that offer battery-free real-time blood pressure sensing and occlusion monitoring ability. The high-pressure sensitivity and the ability to detect subtle changes in vessel motion patterns enable the early detection of partial occlusion (e.g., thrombosis). This work demonstrates a promising approach for incorporating multifunctionality into artificial biological systems for smart healthcare systems [149]. Ganti et al. investigated a watch-based wearable device to determine the noninvasive pulse-transit time-based BP measurements in an at-home setting. The findings of the study suggest that watch-based wearable devices serve as convenient tools to capture migrant BP measurements as compared to traditional oscillometer cuffs. BP values can be determined in real-life practice scenarios with a minimum number of calibration points using this proposed technique [150].

Xuan et al. developed an extremely affordable cuffless BP monitoring device, BPClip, which is compatible with any type of mobile device that has a camera and a flash. This device is of crucial importance to the cost and accuracy of blood pressure measurements. Various design considerations were investigated during the development of BPClip to determine the most intuitive and accessible design features. This work elucidates the blueprint procedure for the development of BP monitoring devices, which can help future developers build accessible health monitoring tools without dependence on existing clinical infrastructure [155]. Shar et al. developed a one-part, flexible, highly conductive, 3D-printable carbon nanotube (CNT)–silicone composite with superior electrical and mechanical properties for advanced health monitoring. Proof-of-concept wearable biosensor devices were investigated in their work for the fabrication of single-component 3D-printed constructs that can interact directly with human tissues. Furthermore, they can also be used for tissue engineering applications, in tactile sensors, health monitoring devices, and as fundamental blocks for flexible biomimetic cell culture platforms [152]. Park et al. fabricated a wireless pressure sensor integrated with a 3D-printed biocompatible polymer stent to record real-time biological signals for smart health monitoring applications. The wireless pressure sensor was fabricated using the MEMS technique. Their study indicates that the anticipated wireless pressure sensor and smart stent will facilitate the development of next-generation biomedical devices for smart health monitoring applications [156]. A smart blood pressure monitor system known as Blood Pressure-ExerGuide was developed by Lin et al. to provide older adults using a community smart gym with customized well-being recommendations. RFID sensors were installed on each piece of smart gym equipment, which enables users to record their BP measurements by using their unique RFID cards. The BP levels of older individuals can be monitored using the Blood Pressure-ExerGuide system, and this system allows for a personalized and secure workout experience. With this system, older individuals can perform exercises on a systematic basis, which further enhances their health and overall well-being [157]. Young et al. developed a 3D-printed ring sensor that encompasses a MEMS piezo-resistive pressure sensor to monitor human BP waveform instantaneously. Using this sensor, the heart rate (HR) and heart rate variability (HRV) can be detected with a precision rate close to that of a commercial EKG chest band. The cardiovascular health status of an individual can be determined using this ring sensor. The features showcased in this research study highlight the potential to monitor BP pulse waveforms in the long term. This sensor can identify changes in waveform properties and mild abnormalities in the initial stages, which in turn enables a comprehensive diagnosis and efficacious precautionary measures [158].

### 4.4. Glucose Biosensors

The application of 3D bioprinting technology has led to the development of glucose biosensors (Figure 8), which can continuously monitor glucose levels in individuals with diabetes. These biosensors, which are printed using additive manufacturing, can be seamlessly integrated into devices and offer a less invasive alternative to traditional methods of monitoring glucose [159,160,161,162,163,164,165]. Nesaei et al. introduced an approach for manufacturing glucose biosensors by modifying commercial carbon ink with Prussian blue as an electron transfer mediator and utilizing a custom-made enzyme ink for 3D printing on tattoo paper using the DIW method. The printed sensors exhibited advantages over conventional screen-printing methods, including enhanced sensitivity, specificity, and reduced material consumption [162]. In another study, Ma et al. developed an enzymatic glucose sensor by combining flower α Ni(OH)2, AuNPs (gold nanoparticles), and β rGO (reduced graphene oxide) to fabricate a nanocomposite with a unique three-dimensional structure. The sensor was fabricated by drop-casting the nanocomposite onto a glass carbon electrode, resulting in sensitivity levels of 559.314 μA mM^−1^ cm^−2^ across a low concentration range and 327.199 μA mM^−1^ cm^−2^ across a high concentration range [166].

Calabria et al. introduced a biosensor that uses a smartphone and 3D printing technology to measure glucose, in real-world samples without the need for chemicals. This groundbreaking device is a significant development in affordable analytical systems based on electrochemiluminescence (ECL), offering applications beyond just measuring glucose levels [167]. Wei et al. presented a hydrogel-based photonic device using a DLP micro-3D printing process for high-sensitivity glucose sensing. Concanavalin A (Con A) hydrogel which is UV-curable was utilized as the sensing material in this work. The results of the study illustrated that the 3D-printed device had an adequate linear response, with a response time of 10–12 min and sensitivity of 0.206 nm/mM. The device has a significant potential for application in real-time glucose monitoring with enhanced performance [171]. Lee et al. developed a glucose-sensing device coupled with a liver-on-a-chip (LOC) platform. Conductive PLA-based three-electrode devices were fabricated via fused deposition modeling using the 3D printing technique. To improve the sensitivity of the tools, multiwalled carbon nanotubes were used [172]. Adams et al. reported a proof-of-concept design and verification of a glucose biosensor fabricated using 3D printing technology. This cutting-edge research work demonstrated the development of a 3D-printed glucose dehydrogenase sensor that meets industry standards. This device can have a significant impact on diabetic treatment as it has proven effective in glucose sensing [173]. For measuring glucose, Silva et al. developed a technique using amperometry. This method resulted in an effortlessly disposable electrochemical sensor made with a sheet consisting of carbon nanotubes (MWCNTs) and nickel oxyhydroxide. The potential for glucose testing in this research is especially remarkable because of the production technique’s affordability and compatibility with conductive surfaces [174].

### 4.5. Oxygen Saturation (SpO2) Biosensors

With the bioprinted oxygen saturation sensors (Figure 9), the saturation levels of oxygen can be monitored in the most reliable way. Integrating these sensors into fingertip devices or textile-based sensors can improve comfort while monitoring for long periods [175,176,177,178,179]. Remote monitoring and the control of oxygen concentrations in 3D cell cultures can be attained using the system proposed by Rivera et al. This system is based on phosphorescence (iPOB), including a chamber for cell culture gas exchange and an integrated oxygen biosensor. This method enables us to remotely monitor oxygen concentrations in 3D tissue constructs, which can be manipulated as needed to encapsulate healthy cell environments [180]. To track the progress of chronic wounds, different biosensors are used. Youseff et al. provided a review of developments made in biosensors and examined the existing limitations and challenges of these biosensors in clinical settings. Furthermore, the paper featured the potential areas of research to address the limitations associated with monitoring wound status using currently available wearable biosensors [175].

The combination of PDMS (polydimethylsiloxane) with electronics is possible by using 3D printing. Abdollahi et al. employed this technique to develop a pulse oximeter called P3 wearable. The intended objective was to fabricate a PDMS cuff through freeform embedding (FRE) printing to achieve a seamless fit for the finger or toe. The feasibility of manufacturing a device that accurately measures blood oxygenation and pulse rate using the optical detection of volume changes in blood vessels is demonstrated through the implementation of this innovative approach [182]. In the healthcare industry, it is crucial to have continuous monitoring capability in emergencies. Contardi et al. investigated the use of MAX30102, a commercial photometric biosensing module coupled to an ESP32 system-on-a-chip device. The Internet of Things capabilities of the modules was also explored in terms of capturing heart rates and oxygen levels from users in real time. The results of device performance aligned with prior research on biosensing medical devices [181]. Zirath et al. conducted a study to investigate how cell numbers, surface coatings of the matrix (ECM) of various types of cells, and the oxygen permeability of chip materials affect the monitoring of oxygen levels during both 2D and 3D cell cultivation. Their research presented an on-chip integrated method for sensing oxygen levels, which is ideal for organ-on-a-chip systems. This method allows for the invasive real-time monitoring of oxygen demands, metabolic activity oxygen uptake rates, and cell viability without the need for labels. Integrating cost-reliable sensor technology into devices holds great potential in understanding physiological or pathological tissue conditions, thus creating new opportunities in biomedical research and pharmaceutical development [184]. Kostecki et al. reported the initial findings from their work on an optical fiber biosensor device that was fabricated using 3D printing technology. This innovative approach resulted in the development of a protein carbonylation biosensor based on fibers. This technology has the potential to offer decision-support information regarding stress levels, benefiting individuals with chronic illnesses, athletes, and livestock animals while also advancing our understanding of factors that influence overall health [185]. An overview of research progress in flexible and wearable all-organic photoplethysmography sensors for monitoring oxygen saturation was reported by Dscosta et al. Comfortable, effective, and reliable pulse oximeter sensors can be achieved for tailored healthcare applications with the amalgamation of novel materials, fabrication techniques, and biocompatible designs [186].

### 4.6. Sweat Biosensors

In recent times, the progress in 3D printing technologies has enabled the fabrication of sweat biosensors to capture different types of biomarkers such as sweat in real time, as shown in Figure 10. These sensors can be utilized to provide insights into electrolyte balance, metabolic markers, and hydration levels; moreover, they can be used to assess athletic performance and for therapeutic diagnostics [187,188,189,190]. Koukouviti et al. developed a 3D-printed sensor modified with a water-stable complex of Fe(III) basic benzoate for the voltammetric detection of glucose (GLU) in acidic conditions of the epidermal skin. The combination of 3D printing technology and the excellent electrochemical performance of drop-casting Fe (III) clusters led to the development of a novel 3D-printed wearable sensor with advanced and cost-efficient enzyme-free sensing ability for noninvasive applications [191]. Kim et al. presented an innovative, flexible, personalized, and cost-effective AIIW: an all-inclusive integrated wearable patch produced with a 3D printing technique. Various electrolyte levels in a person’s sweat can be captured uninterruptedly and noninvasively using this AIIW patch. This study serves as a step closer to the development of a next-generation noninvasive personalized health monitoring system by utilizing the merits of 3D printing techniques [192]. Padash et al. developed a wearable microfluidic gadget for sweat analysis to accumulate sweat from the human skin. Multi-jet modeling (MJM) was utilized for the fabrication of the wearable gadget by using flexible materials. The findings of the study reveal that the gadget can effectively capture sweat analytes in a rapid and timely manner. This microfluidic device can collect an individual’s sweat and analyze its components directly on the skin surface [193].

Wu et al. presented a novel epifluidic (epidermal microfluidic) device with digital light processing. This 3D-printed epifluidic platform, labeled as “sweatainer”, illustrates the capability of 3D design for microfluidics via the production of fluidic components with complex architectures previously considered unfeasible. The sweatainer system can collect multiple, individual sweat samples for either peripheral or on-body analysis. This platform represents a crucial improvement in the collection and assessment of sweat samples [196]. Song et al. illustrated the fabrication of epifluidic elastic electronic skin (e3-skin) with an extrusion-based 3D printing technique. The e3-skin device possesses extraordinary multimodal physiochemical sensing abilities and can detect the concurrent physiological state of individuals when performing regular daily activities. Moreover, they also predicted the reaction time and the inhibitory control of the individual after alcohol intake by using machine learning algorithms. This study paves the path for the potential autonomous fabrication of personalized wearable systems for monitoring health and for use in clinical applications [194]. Weng presented a compact 3D-printed microfluidic origami wearable biosensor based on a smartphone to capture cortisol levels in human sweat. The study results illustrated that the biosensor successfully captured the cortisol levels in a real sweat sample of humans. This portable biosensor enables a rapid, cost-effective, and noninvasive sensing solution for personalized healthcare based on sweat analysis [197]. The above-mentioned research efforts enable the accurate monitoring of diseases and physical conditions of patients who need to be monitored constantly in daily life with advanced healthcare applications [198].

### 4.7. Tactile Biosensors

The integration of tactile sensors into wearable patches or gloves has been made possible by the advances in 3D bioprinting for healthcare applications, as shown in Figure 11. Tactile sensors can capture elements such as touch, pressure, and temperature changes, with application in biomedical devices and prosthetics [199,200,201,202]. H. Li et al. presented a novel fabrication method for developing an all-printed soft triboelectric nanogenerator (TENG) by utilizing the inkjet and direct ink-writing 3D printing technique. The developed TENG sensor has the ability to generate electricity as an energy transfer and multifunctional sensing device. The all-printed TENG has enormous potential to be used as a tactile sensor in energy and smart e-skin applications [203]. Nag et al. presented the design, fabrication, and characterization of PDMS/graphite sensors for sensing applications. Flexible sensor patches were fabricated by initially building the molds onto which PDMS and graphite powder were cast. The flexibility of the 3D printing method allowed the researchers to build the mold in a cost-effective way. This sensor can also be utilized for multiple other sensing applications such as determining temperature, humidity, and gas [204]. Haque et al. demonstrated the fabrication of a vertical-contact separation mode triboelectric nanogenerator with a 3D printing technique by using soft materials. The device operated as a self-power sensor and provided unique electrical responses to the adjustments in operational dynamic area, occurrences, and applied force. This study paves the way for the major advances of digital fabrication in conjunction with 3D smart objects using soft materials for robotics and wearable applications [201]. Qu et al. presented the development of 3D-printed strain-gauge microforce sensors with micronewton sensing resolutions. They used 3D printing techniques such as fused deposition modeling and stereolithography to build polymeric microsensors, and their performance was further analyzed to measure the sensor sensitivity and accuracy. The results of their study illustrate that a customized force-sensing system can be developed with a flexible design using this proposed 3D printing technique [205].

Yi et al. reported all-3D-printed, flexible, and hybrid wearable bioelectronic tactile sensors by using biocompatible nanocomposites for real-time health monitoring applications. These pressure sensors can capture multiple physiological signals such as the respiration, vibration, and radial artery pulse of an individual. This 3D printing approach provides a rapid, novel, and low-cost fabrication method and contributes to advances in smart tactile sensing and human–machine interface applications [76]. Shar et al. investigated the 3D printing of conductive elastomers (one-part carbon nanotube) for customized patient-specific health monitoring applications. A one-part 3D-printable carbon nanotube–silicon composite ink was developed and systematically characterized in this work. This device serves as a proof of concept to establish a one-part 3D-printed component that can interact directly with human organs in advanced health monitoring and bionic skin applications [152]. Presti et al. presented a 3D-printed tactile sensor for breast cancer identification based on fiber Bragg grating (FBG) technology. This study addresses prior issues such as the low force sensitivity and high power consumption of tactile sensors. The findings of their study provide guidance to clinicians or self-users for detecting breast tissue tumors noninvasively in an effective way [207].

### 4.8. Respiratory Biosensors

The design flexibility and development of novel materials have advanced the 3D printing methods used to manufacture respiratory biosensors for monitoring breathing patterns, respiratory rate, virus detection, and lung function, as illustrated in Figure 12. Wearable biosensors have the capability to collect valuable data and provide crucial insights for the timely detection of respiratory illnesses and the optimization of respiratory remedies [152,208,209,210,211,212,213,214].

Martins et al. demonstrated the fabrication of an immunosensor with fused deposition modeling using a conductive filament of carbon black and polylactic acid (PLA). In their study, for the first time, the construction of a commercial 3D conductive filament of carbon black and polylactic acid (PLA) was reported, which was used to detect Hantavirus Araucaria nucleoprotein (Np) as a proof of concept. This immunosensor platform can also be utilized for clinical diagnostics such as the biosensing of other electrochemical signals [213]. Jodat et al. developed a hybrid dual bioink-printed nose device with an integrated biosensing mechanism. The 3D cartilage-like tissue constructs were fabricated by mechanically fine-tuning the chondrocyte-laden bioink materials using a multimaterial 3D bioprinting process. Furthermore, the functional olfactory sensations were realized by integrating a biosensing system into the engineering tissue construct. This study provides a foundation for humanoid cyborgs and functional bionic interfaces [217]. Chen et al. presented a flexible humidity sensor fabrication by using a direct-writing 3D printing process. The materials used were graphene nanoflakes (GNPs)/multiwall carbon nanotubes (MWCNTs) as the sensor element and polydimethylsiloxane (PDMS) as the substrate. The findings of their study reveal that the fabricated sensor has superior properties such as excellent flexibility, reliable stability, high sensitivity, high repetitiveness, and ultra-fast response and recovery times. This sensor has great potential to track respiratory moisture levels for the effective clinical diagnosis and treatment of respiratory issues [215]. Mustafa et al. developed a customized 3D-printed breath analyzer by integrating cerium oxide nanoparticles for colorimetric enzyme-based ethanol sensing. The alcohol level in the user’s breath can be determined when they blow their breath on the sensor surface. According to their study, it is one of the first reported 3D-printed colorimetric biosensors based on enzymes for breath analysis. Moreover, the findings illustrate that this biosensor is able to measure the concentrations of breath ethanol at levels equivalent to that of the electronic breathalyzers, with superior sensitivity and low cost. This device is simple to operate, sensitive, and portable and can be used to determine ethanol in various sectors such as forensics, the food and beverage industry, and law enforcement [216]. Panahi et al. fabricated a porous pressure sensor with a cone structure for wearable respiratory monitoring applications. This pressure sensor consists of three components: a master mold, a dielectric layer, and fabric-based electrodes. The master mold was fabricated with a 3D printing process; a dielectric layer was formed by annealing a mixture of nitric acid (HNO3), PDMS, and sodium bicarbonate (NaHCO3) in the master mold; and the electrodes were generated by screen printing silver on fabric. The porous and cone structures exhibited remarkable deformation. This sensor has high sensitivity, with ≈ 530% kPa^−1^ for ultra-low-pressure ranges below 10 pascals. The findings indicated that the cone-structure pressure sensor attached to the mask detected discrete respiration rates of humans [218].

### 4.9. A Comparison of 3D-Printed Wearable Biosensors with Traditional Wearable Biosensors

A comparison of 3D-printed biosensors with traditional (TRD) biosensors is provided in Table 4. Based on the findings reported by various researchers, shown in Table 4, it can be inferred that 3D-printed biosensors enable the detection of biomarkers with high sensitivity compared to traditional biosensors.

## 5. Challenges

The swift progress in 3D printing and nanomaterials has revolutionized the scope of wearable biosensors for point-of-care testing, health monitoring, and clinical diagnostics. Even though significant developments have been achieved in 3D-printed wearable biosensors, there are certain key issues. The numerous technical issues hindering the full-scale development of 3D-printed wearable sensors include material standards, sustainability, stability, quality, wearability, power consumption, privacy, and security vulnerabilities, which are shown in Figure 13.

Material standards: So far, there are no international standards developed on the selection of materials for 3D-printed wearable biosensors used in medical applications. The safety and efficacy of the materials need to be determined before using them for 3D printing purposes. Hence, comprehensive material standards for medical device fabrication should be developed by regulatory agencies to fully adopt the versatility of 3D printing processes [101,219].

Health risks and sustainability concerns: The use of nanocomposites in 3D-printed biosensors could be a serious issue. Due to the small size of the nanoparticles, they might pass through the human cell membrane while capturing the data over a long period. From this perspective, health and safety issues can play a critical role in 3D-printed wearable biosensors used for healthcare applications. Hence, further studies need to be conducted on the impact of nanocomposite materials to build bio-friendly, safe, and sustainable biosensors [56,107].

Stability: In 3D printing, there are numerous process parameters that impact the properties of a 3D-printed wearable biosensor such as sensitivity, quality, stability, and mechanical strength [169]. Sometimes, it might be very difficult to identify the defects caused due to the selection of erroneous process parameters. These defects might lead to inaccurate readings or malfunction of the wearable biosensor in a real-time monitoring scenario, which is a huge risk to patient health. Thus, it is essential to optimize these process parameters with real-time closed-loop control to develop reliable and durable biosensors for health monitoring applications [56].

Quality: The surface contamination of the sensor can have an enormous effect on the accuracy of a 3D-printed wearable sensor’s measurements. When these biosensors are exposed for a long period of time to track data in an intense environment, it can lead to potential surface contamination in sensors, signal calibration issues, and inaccurate measurements. By using dynamic calibration techniques such as multimarker sensing and drift correction, or applying a strong antipollution sensor surface coating, the long-term reliability of 3D-printed wearable sensors can be enhanced [101].

Wearability: The most common setback is the wearability of 3D-printed biosensors, which refers to their lifecycle. If the device has short durability or lifespan, then it leads to increased consumption of these devices, and users would have to buy new devices quite frequently to maintain functionality. This in turn leads to more wastage of electronic devices. So, to develop 3D wearable biosensors with an enhanced lifecycle, every production step must be considered for long-term performance [58,162].

Power consumption: This issue can be addressed by utilizing energy storage components and self-powered devices to ensure that 3D-printed wearable biosensors are supplied with the necessary power to monitor human health parameters continuously without any interruptions. In addition, minimizing circuit power consumption and harnessing power from patient movements can extend the life of batteries for wearable devices. Energy can be harvested from humans or the ambient environment to prolong the working life of wearable devices [220]. Inertial systems have been used to generate electricity from extreme movements using hands, legs, and other body parts [221]. Human-centric energy harvesting methods can range from biochemical to biomechanical methods. Environment-centric energy harvesting includes infrared radiation, radio-frequency signals, and solar energy, to name a few [222]. Furthermore, a hybrid approach can be implemented with a combination of the above-mentioned technologies to generate sustained levels of electric power. Microscale 3D printing can aid in the fabrication of microelectromechanical (MEM) devices, which have been traditionally carried out using lithography-based methods [223]. In addition, 3D-printed microscale techniques can enable the development of hierarchical multilayered structures as well as conformal [224] conductive traces for compact devices that can be contained within the wearable device footprint.

Privacy and security vulnerabilities: Data collection, transmission, and protection are some of the major challenges of wireless wearable biosensors. A patient’s personal health information collected via sensors can be stolen or manipulated by hackers to engage in criminal and illegal activities. It is very crucial to determine a trustworthy interface mechanism to ensure accurate data acquisition. The health status of another individual can be falsified as that of the patient or vice versa, and in both situations, this may put an individual’s life at risk. Due to the lack of features such as privacy considerations and data security, wearable biodevices are prone to intrusive hazards due to prospective cyberattacks [225,226]. Thus, it is very crucial to develop data processing protocols and intrusion mitigation techniques to predict and protect from possible hazards [101].

## 6. Prospects

Customized healthcare medicine is expected to have a huge demand in the future. By utilizing the design capabilities and material flexibility of 3D printing processes, personalized robust point-of-care treatment strategies and skin wearable biosensors can be established, as shown in Figure 14. Numerous ongoing studies are being conducted by researchers and wearable devices industries to advance 3D multimaterial printing and production scale-up to produce biosensors with controlled texture and topography. Currently, there are only a limited number of biomarkers assessed by 3D-printed wearable biosensors. It is crucial to understand the components of each organism’s bodily fluid and blood chemistry for the widespread adoption of wearable biosensors. With the advancement in sensing mechanisms and fabrication techniques, future wearable biosensors may be able to capture all physiological markers noninvasively for therapeutic and comprehensive clinical diagnosis. Furthermore, these devices can monitor health indicators continuously in real time and provide the respective guidelines for personalized treatment. Building upon 3D printing methods, 4D printing is a novel field in which the shape, physical properties, and functionality of 3D-printed wearable biosensor constructs can be transformed as a function of time by triggering the sensors. This can be achieved by applying an external stimulus such as optical irradiation, temperature fluctuation, or chemical reaction. Notably, 4D printing technology advances the self-active sensing ability of wearable biosensors for real-time physiological monitoring for self-diagnosis. Using intelligent and responsive materials in 4D printing processes enables researchers to develop dynamic structures with revolutionary potential across the healthcare field.

With state-of-the-art AI and cloud computing technologies, the recorded biosensor data can be transmitted directly to the respective clinician remotely to provide real-time clinical diagnosis. Many factors such as the reproducibility of sensor production, mechanical interference, biofouling issues, and shelf life still need further research for the structural optimization of wearable devices and material engineering. With the aid of artificial intelligence (AI) and machine learning (ML) technology, various wearable biosensors can generate specific automated responses to biomarkers, and the compositions of the tested fluids can be interpreted successfully for advanced clinical diagnostic applications. Furthermore, AI can also be utilized to enhance the self-calibration abilities of WBSs for the verification of sensing accuracy and the realization of calibration-free biosensors for consistent real-world health monitoring. These pioneering studies of wearable biosensors pave the path toward the development of smart wearable biosensors for reliable point-of-care applications, which is one of the fundamental aims of biosensors.

## 7. Conclusions

With the progress in 3D printing technologies and novel material development, researchers on wearable devices have drastically shifted their focus from just tracking individual exercise routines to addressing crucial healthcare issues such as geriatric remote monitoring and clinical diagnosis for chronic diseases. This review outlines the significant developments of 3D printing processes in the past decade for manufacturing novel next-generation wearable biosensor constructs using nanocomposite materials. The fabrication of biosensors using 3D printing technology is a superior approach compared to conventional manufacturing processes. More specifically, we reported on the advances for capturing and monitoring biomarker elements such as heartbeat, blood pressure, brain waves, body temperature, oxygen saturation, glucose levels, and breathing patterns. Finally, we discussed the potential technical obstacles for 3D-printed wearable biosensors and their prospects for facilitating advanced healthcare and enhancing human well-being.

## Figures and Tables

**Figure 1 bioengineering-11-00032-f001:**
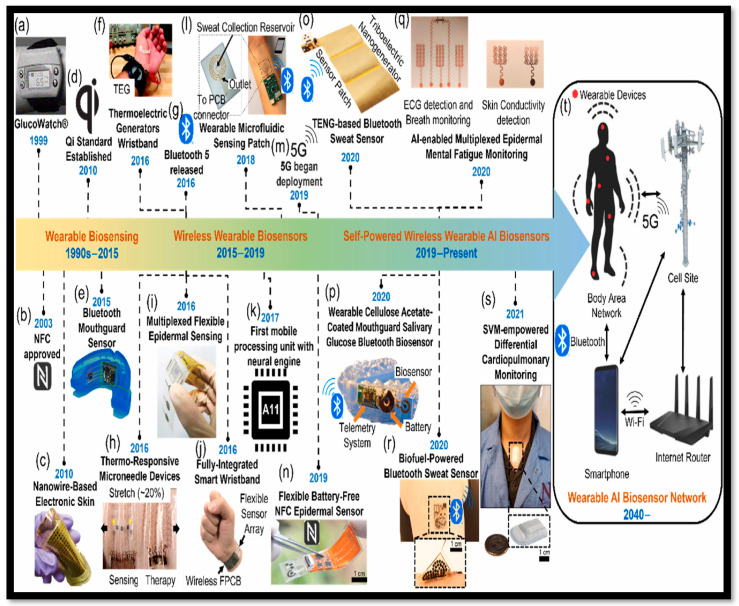
The development timeline of wearable biosensor devices: (**a**) GlucoWatch; (**b**) NFC-approved sensors; (**c**) nanowire-based electronic skin; (**d**) Qi standard established sensors; (**e**) Bluetooth mouthguard sensors; (**f**) TEG (thermoelectric generator) wristbands; (**g**) Bluetooth 5 sensors; (**h**) thermo-responsive microneedle devices; (**i**) multiplexed flexible epidermal sensors; (**j**) fully integrated smart wristbands; (**k**) first mobile processing unit with a neural engine; (**l**) sweat collection reservoir, wearable microfluidic sensing patch; (**m**) 5G began deployment; (**n**) flexible battery-free NFC epidermal sensors; (**o**) TENG-based Bluetooth sweat sensors; (**p**) wearable salivary glucose Bluetooth biosensors; (**q**) AI-enabled multiplexed epidermal mental-fatigue-monitoring biosensors; (**r**) biofuel-powered Bluetooth sweat sensors; (**s**) SVM-empowered differential cardiopulmonary monitoring sensors; (**t**) AI-enabled biosensors, © 2022 Elsevier [28].

**Figure 2 bioengineering-11-00032-f002:**
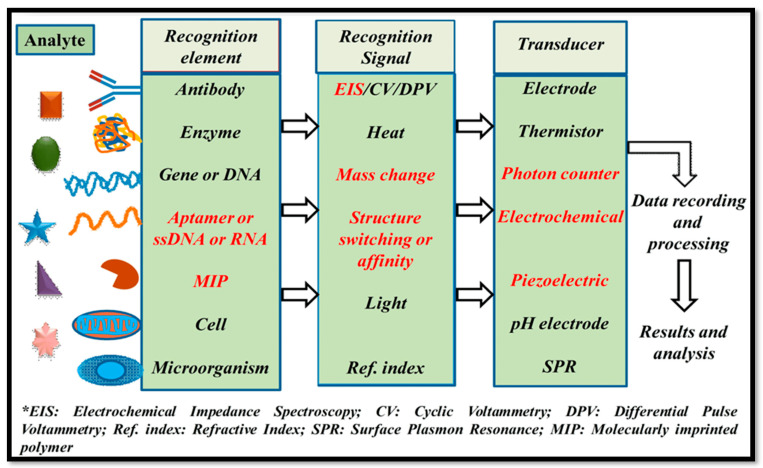
The schematic interpretation of the biosensor components [2]. The asterisk (*) represents only the full forms of the abbreviations (EIS, CV, DPV, Ref. index) & (SPR) & (MIP) listed inside the blocks (Recognition Signal, Transducer, and Recognition Element) respectively for Figure 2.

**Figure 3 bioengineering-11-00032-f003:**
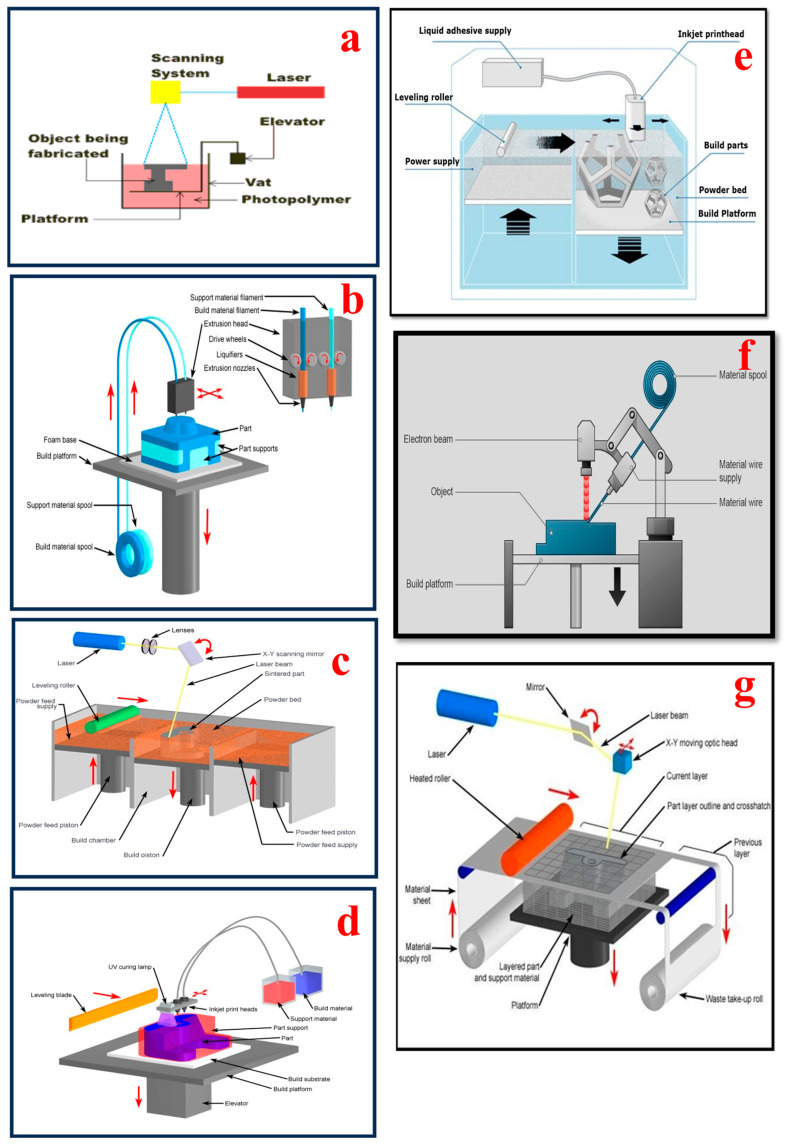
Additive Manufacturing Processes: (**a**) Vat photopolymerization (VP); (**b**) extrusion-based system (EBSs); (**c**) powder bed fusion (PBF); (**d**) material jetting (MJ); (**e**) binder jetting (BJ); (**f**) directed energy deposition (DED); (**g**) sheet lamination process (SLP) [96].

**Figure 4 bioengineering-11-00032-f004:**
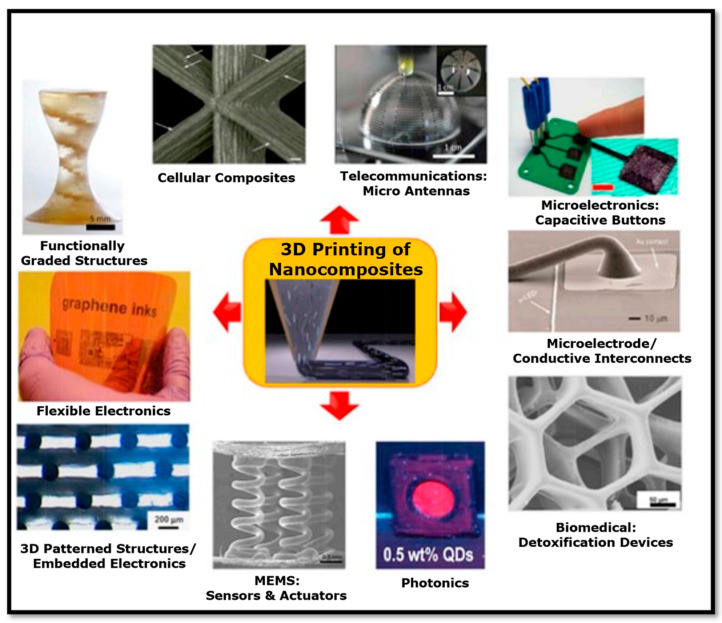
Images of 3D-printed nanocomposites for a wide range of applications, such as MEMS, photonics, microfluidics, biomedical devices, microelectronics, and telecommunication tools [94].

**Figure 5 bioengineering-11-00032-f005:**
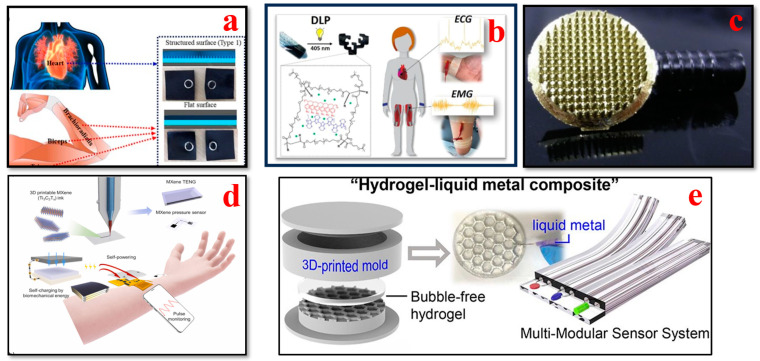
Images of 3D-printed electrocardiogram (ECG) biosensors: (**a**) 3D-printed sensing electrodes attached to skin © 2023 Elsevier Ltd. [121]; (**b**) DLP-printed conductive hydrogel © 2011 Elsevier Ltd. [126]; (**c**) 3D-printed dry electrode © 2022 Creative Commons-CC-BY 4.0. [122]; (**d**) MXene-based self-powered physiological sensing system © 2020 Creative Commons-CC-BY 4.0 [59]; (**e**) hydrogel–liquid metal composite © 2022 Elsevier Ltd. [127].

**Figure 6 bioengineering-11-00032-f006:**
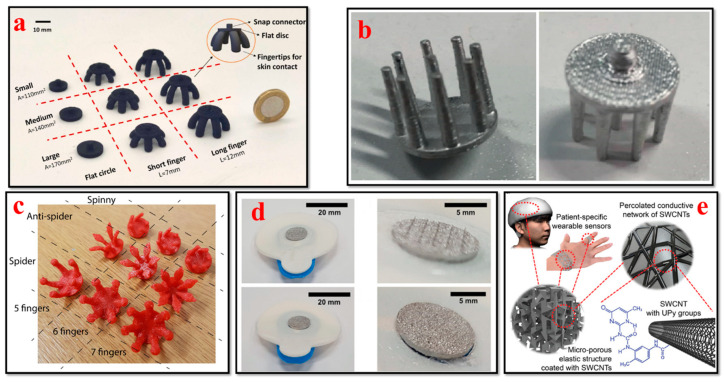
(**a**) Different electrode designs in 3 categories: flat circle, short-fingered, and long-fingered © 2022 Creative Commons CC-BY [136]; (**b**) 3D-printed EEG electrode coated with silver paint © 2016 MDPI [137]; (**c**) nine different 3D-printed electrode configurations © 2019 Velcescu et al. Creative Commons CC-BY [138]; (**d**) 3D-printed microneedle and flat electrode © 2020 Wiley-VCH GmbH [139]; (**e**) schematic of the hierarchical structure of a 3D-printed object and conductive network © 2019 Creative Commons CC-BY [140].

**Figure 7 bioengineering-11-00032-f007:**
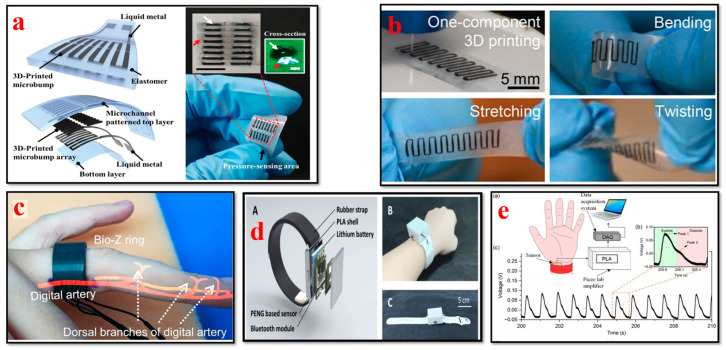
(**a**) Schematic view of proposed and fabricated 3D-printed rigid micro-bump-integrated liquid metal-based pressure sensor (3D-BLiPS) © 2019 Wiley-VCH Verlag GmbH & Co. KGaA, Weinheim [151]; (**b**) a 3D-printable highly conductive, flexible, stretchable one-part CNT–silicone ink © 2022 Wiley-VCH GmbH [152]; (**c**) a 3D-printed bioimpedance (Bio-Z) ring sensor for arterial blood flow and pressure sensing © 2023 Creative Commons CC BY 4.0 [148]; (**d**) The overview of Blood pressure predict wristband (BBPW): (A) The structure of BPPW and the materials used in each part. (B) The photograph of subjects wearing BPPW. (C) The photograph of BPPW; the whole length of BPPW is 26 cm © 2022 Creative Commons CC BY 4.0 [153]; (**e**) (a) Schematic illustrating the process of pulse pressure waveform (PPW) collection using the 3D-printed sensor. (b) Peak 1 and peak 2 represent systolic pressure and diastolic pressures, respectively. (c) PPW captured from a human subject. © 2023 Wiley-VCH GmbH [154].

**Figure 8 bioengineering-11-00032-f008:**
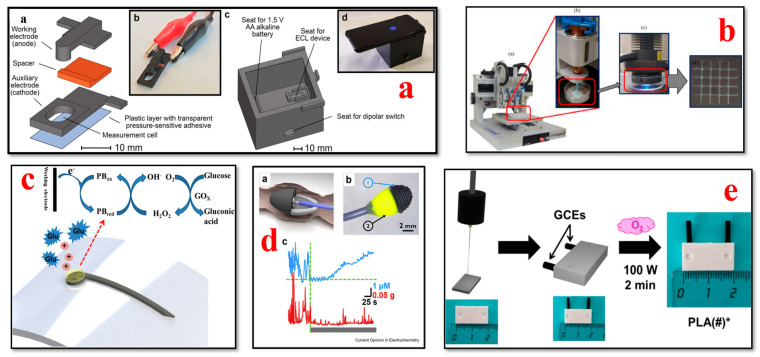
(**a**) a. Scheme and b. photograph of the 3D-printed electrochemiluminescence enzyme biosensor device. c. Scheme of the 3D-printed dark box d. photograph of the analytical system during a measurement © 2023 Calabria et al., published by Elsevier B.V [167]; (**b**) 3D-printed workflow of glucose biosensor: a. bioprinter, b. printing of sensor ink onto plasma-treated Petri dish, c. UV curing, d. printed structure © 2023 Krstić et al., published by Elsevier Masson SAS [168]; (**c**) a 3D-printed biosensor for glucose sensing © 2018 Elsevier B.V. All rights reserved [162]; (**d**) a. Schematic visualization of the electrochemical sensor, showcasing, b. Final 3D-printed electrochemical sensor, where (1) is the carbon black/PLA electrode and (2) is the PLA sealing cap, c. Simultaneous monitoring of 5-HT overflow and circular contraction from the anorectum © 2020 Elsevier B.V. All rights reserved [169]; (**e**) 3D-printed poly(lactic acid) electrochemical multisensors © 2023 Escartín et al., published by Elsevier B.V. [170] (The chemically modified prototype, which acted as working electrode in the sensing process, was denoted PLA(#)* where # indicates the pressure of O_2_ used during the process that transforms the electrochemically inert PLA into an electrochemically active material).

**Figure 9 bioengineering-11-00032-f009:**
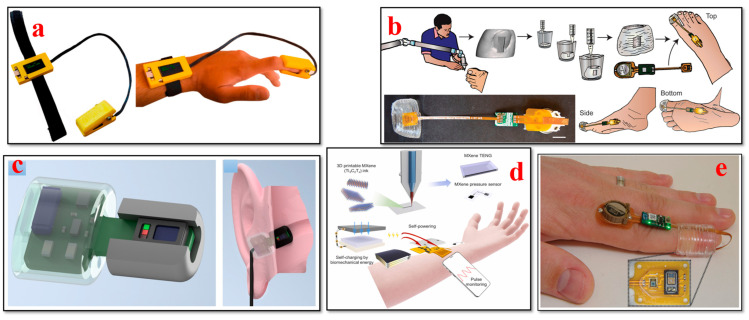
(**a**) Photometric biosensing module integrated with 3D-printed thermoplastic polyester case © 2022 Creative Commons Attribution CC BY 4.0 [181]; (**b**) 3D-printed toe-cuff sensor with PDMS using FRE technique © 2020 WILEY-VCH Verlag GmbH & Co. KGaA, Weinheim [182]; (**c**) a wearable in-ear photoplethysmography sensor and a 3D-printed case to house the circuitry © Creative Commons Attribution CC BY 4.0 [179]; (**d**) schematic of the MXene-based self-powered physiological sensing system (MSP^2^S^3^), which comprises a MXene-based TENG (M-TENG) and a MXene-based pressure sensor (M-PS), both fabricated via additive manufacturing (3D printing) © 2022 Elsevier Ltd. All rights reserved [183]; (**e**) 3D-printed wearable pulse oximeter for the finger © 2020 WILEY-VCH Verlag GmbH & Co. KGaA, Weinheim [182].

**Figure 10 bioengineering-11-00032-f010:**
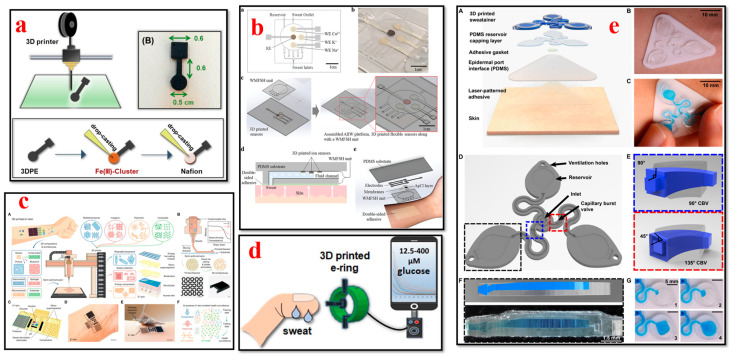
(**a**) Schematic of the fabrication process and the 3D-printed voltammetric sweat sensor (B) modified with a Fe(III) cluster © 2022 Creative Commons Attribution CC BY 4.0 [191]; (**b**) schematic illustration of an all-inclusive integrated wearable (AIIW) 3D-printed wearable bioelectronic patch: a. Top-view schematic showing the composition of the AIIW patch. b. Optical image of a flexible AIIW patch, c. The individual components of the AIIW patch, d. Cross-section of the AIIW patch, e. The individual components of the AIIW patch © 2021 Wiley-VCH GmbH [192]; (**c**) semisolid extrusion (SSE)-based 3D-printed e3-skin: A. Schematic of the SSE-based 3D printing, B. Schematic illustration of SSE printing procedures to prepare 2D and 3D architectures, C. 3D-printed e3-skin, D and E. Optical images of an e3-skin, F. Machine learning–powered multimodal e3-skin © 2023 Song et al. Creative Commons Attribution License 4.0 (CC BY) [194]; (**d**) 3D-printed electrochemical ring for sweat analysis © 2021 American Chemical Society [195]; (**e**) A. An exploded render highlights key components of the sweatainer system, B. The sweatainer mounted on the ventral forearm of an individual before the onset of sweat collection, C. The construct of the sweatainer eliminates uncontrolled fluid transport under mechanical loading, D. Illustration of the sweatainer highlighting key device, E. Renders of three-dimensional (3D) CBV designs, F. CAD render (top) and photograph of actual device (bottom), G. Photographic sequence highlighting the complete filling of a sweat collection reservoir. © 2022 Wu et al. Creative Commons Attribution License 4.0 (CC BY) [196].

**Figure 11 bioengineering-11-00032-f011:**
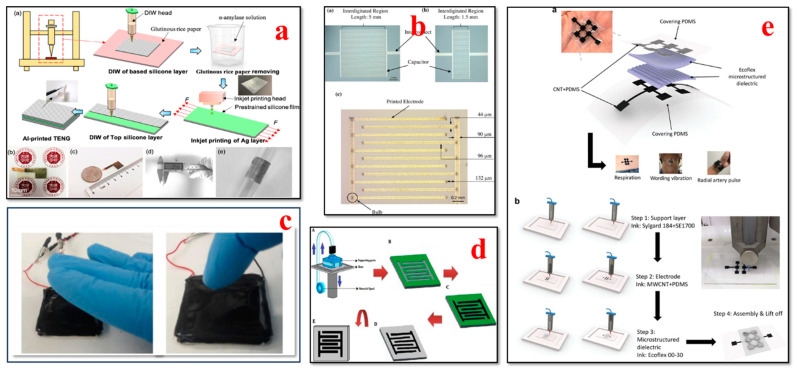
(**a**) Fabrication steps of an all-printed triboelectric nanogenerator for tactile sensing © 2020 Elsevier Ltd. All rights reserved [203]; (**b**) Aerosol jet 3D-printed capacitive sensor: a. electrode length of 5 mm, b. electrode length of 1.5 mm, and c. high magnification image of the sensor in b © 2016 Elsevier B.V. All rights reserved [206]; (**c**) A self-powered triboelectric touch sensor made of 3D-printed materials © 2018 Elsevier Ltd. All rights reserved [201]; (**d**) 3D-printed mold-based graphite/PDMS sensor for tactile sensing: A. 3D printing resulted in the reusable mould, B. Then, graphite powder was cast, C. onto the mould, filling its trenches. This was followed by the casting of PDMS D, which was cured to form the sensor patches E. © 2018 Elsevier B.V. All rights reserved [204]; (**e**) Schematic of flexible wearable tactile sensors: a. Photograph and the schematic of the exploded view of the M2A3DNC pressure sensor for multiple physiological signals monitoring, b. A schematic representation of 3D printing of inks © 2021 Wiley-VCH GmbH [76].

**Figure 12 bioengineering-11-00032-f012:**
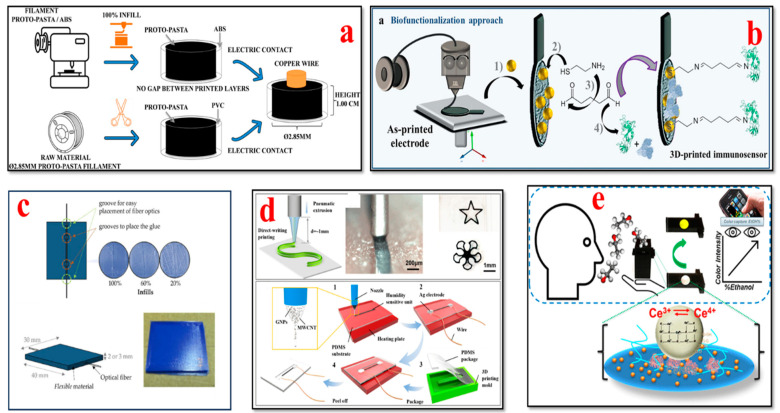
(**a**) A schematic of 3D-printed electrode as a new platform for electrochemical immunosensor © 2020 Elsevier B.V. All rights reserved [213]; (**b**) 3D-printed electrochemical COVID-19 immunosensor, fabrication steps © 2021 Elsevier B.V. All rights reserved [214]; (**c**) 3D-printed sensor based on fiber Bragg grating (FBG) technology for respiratory rate (RR) and heart rate (HR) monitoring © 2022 Optica Publishing Group [211]; (**d**) a fast-response non-contact flexible humidity sensor: Top- direct write inkjet printing principle followed by liquid ajection and the optical images of different patterned structures, Bottom- Manufacturing process flow of the flexible humidity sensor © 2023 Chen et al. Creative Commons Attribution License 4.0 (CC BY) [215]; (**e**) a 3D-printed breath analyzer incorporating CeO2 nanoparticles © 2021 American Chemical Society [216].

**Figure 13 bioengineering-11-00032-f013:**
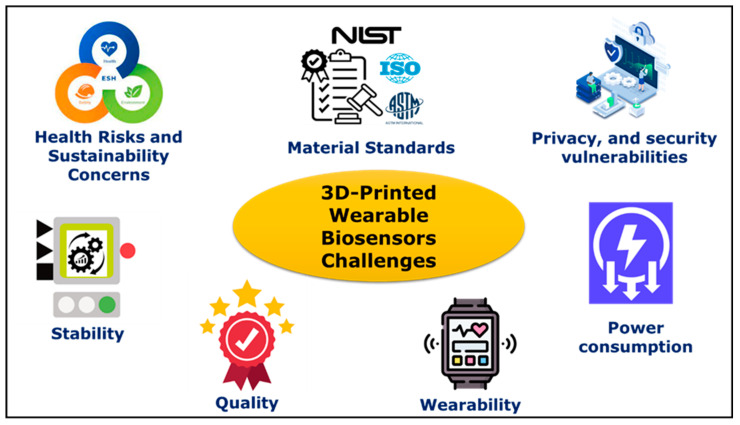
Schematic of 3D-printed wearable biosensor challenges.

**Figure 14 bioengineering-11-00032-f014:**
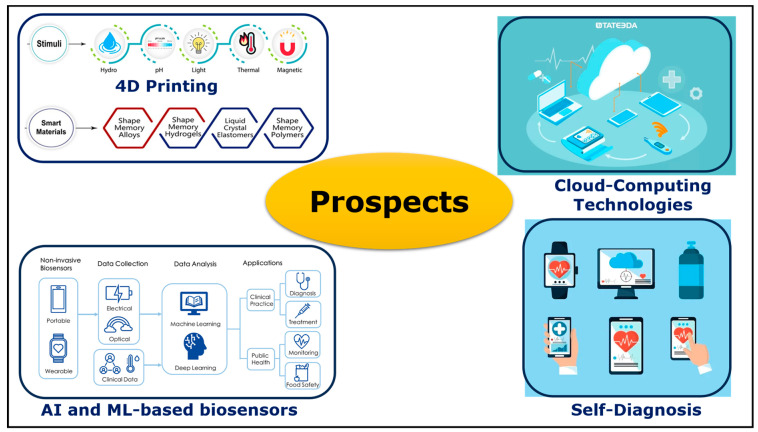
Schematic representing the prospects of the wearable biosensors © 2023 Commons Attribution License 4, © 2021 Wiley-VCH GmbH [227,228].

**Table 1 bioengineering-11-00032-t001:** Some of the commercially available wearable biosensors [24].

Company	Model	Analyte	Measuring Range (mM)
Yellow Springs Instruments, Yellow Springs, OH, USA	23 A 13 L	Glucose Lactate	1–45 0–15
Zentrum für Wissenschaftlichen Geratebau, Berlin, Germany	Gluco- meter	Glucose Uric acid	0.5–50 0.1–1.2
Abbott, Abbott Park, Illinois, USA	FreeStyle InsuLinx	Glucose, insulin	20–500 mg/dL
Lifestream cholesterol monitor Alcosan saliva alcohol dipstick	i-STAT PCA	Glucose Urea nitrogen, Cl, K, Na^+^, hematocrit blood gases	20–700 mg/dL -
DEX blood glucose meter	Bio-scanner 2000	Glucose, cholesterol, HDL, blood ketone, triglyceride	55–230 mg/dL -
Germaine Laboratories, Inc., San Antonio, TX, USA	AimStrip hemoglobin meter	Hemoglobin	5.6 to 23.5 g/dL

**Table 2 bioengineering-11-00032-t002:** Different types of nanocomposites [90].

Class	Examples	Properties
Polymer	Thermoplastics/layered silicates/thermoset polymers, polymer/CNT, polyester/TiO_2_, polymer/layered double hydroxides.	Enhanced electrical conductivity and colloidal stability, biodegradability
Ceramic	Al_2_O_3_/CNT Polymer, Al_2_O_3_/TiO_2_, SiO_2_/Ni, Al_2_O_3_/SiC, Al_2_O_3_/SiO_2_	High toughness and superior failure properties
Metal	Fe-Cr/Al_2_O_3_, Al/CNT, Mg/CNT, Co/Cr, Fe/MgO, Ni/Al_2_O_3_	Strong ductility and excellent shear/compression practices

**Table 3 bioengineering-11-00032-t003:** Overview of AM processes [102,103].

Process	Technology	Materials	Minimum Layer Resolution	Max Build Volume (LxWxH-mm^3^) and Applications
Vat photopolymerization	Stereolithography (SLA) Digital light processing (DLP) Continuous liquid interface production (CLIP) Scan, spin, and selectively photocure (3SP)	Photopolymers	50–100 µm 25–150 µm 50–100 µm 25–100 µm	1500 × 750 × 550 192 × 120 × 230 190 × 112 × 325 266 × 175 × 193 Rapid prototypes, tooling, end-user parts, and mold patterns.
Extrusion-based systems	Fused deposition modeling (FDM)	Thermoplastics (PLA, ABS, HIPS, Nylon, PC)	10–100 µm	1500 × 1100 × 1500 Spare parts, automotive, testing tool designs, and jigs
Powder bed fusion	Selective laser sintering (SLS) Electron beam melting (EBM) Selective laser melting (SLM) Selective heat sintering (SHS) Direct metal laser sintering (DMLS)	Polymers, metals and ceramic powder	80 µm 70 µm 20–50 µm 100 µm 20–40 µm	381 × 330 × 460 6096 × 1194 × 1524 300 × 300 × 300 160 × 140 × 150 250 × 250 × 325 Aerospace, automotive, dental, rapid prototyping, and jewelry
Material jetting	Multi-jet modeling, drop-on-demand, thermo-jet printing, and inkjet printing	Polymers, plastics, and waxes	13 µm	300 × 185 × 200 Casting patterns, prototypes, and electronics
Binder jetting	3D printing	Polymers, waxes, metals, and foundry sand	90 µm	2200 × 1200 × 600 Prototypes, casting patterns, and molds
Directed energy deposition	Laser engineering net shape (LENS)	Metals	50–100 µm	1500 × 1500 × 2100 Aerospace, military, repair metal objects and satellites
Sheet lamination processes	Laminated Object manufacturing (LOM)	Metals, paper, plastic film	100 µm	256 × 169 × 150 Prototypes, plastic parts, and end-user parts

**Table 4 bioengineering-11-00032-t004:** Comparison of 3D-printed biosensors with traditional (TRD) biosensors © 2021 Wiley-VCH GmbH [43].

Methods	Biomarker	Sensor Structures (3D and Non-3D)	Detection Range	LoD	Sensing Capabilities and Remarks
3D printing (SLA)	Prostate-specific antigen	3D-printed channels; immunoarray	0.5 pg mL^−1^ to 5 ng mL^−1^	0.5 pg mL^−1^	Customizability and rapid prototyping capability. Automated detection system and assay time ≈ 30 min. Accuracy comparable with ELISA and commercial devices such as Abbott Diagnostics (0.008 ng mL^−1^), Roche (0.002 ng mL^−1^), Beckman Coulter (0.008 ng mL^−1^), and Diagnostic Products Corporation (0.04 ng mL^−1^)
Commercial SPR biochip (TRD)		Self-assembled monolayered Au	1–1000 ng mL^−1^	18.1 ng mL^−1^	Assay time ≈ 14 min. Sensing with buffer solution and human serum
Microfabrication (TRD)		Self-assembled monolayered Au	0–4 µg mL^−1^	0.2 µg mL^−1^	Single-use biosensor, sensing with serum samples and good sensitivity
3D printing (AJP)	Dopamine	Micropillar array electrode	100 am–1 mm	am	Low LoD ≈ 500 attomoles, breaking the barrier described in the literature [47] through multi-length-scale electrode structure. Rapid prototyping capability and waste minimization due to the small microfluidic volume required for testing.
3D printing (2PP)		3D carbon electrode	0.5–100 µm	nm	High sensitivity to multiple neurochemicals, high reproducibility, and capability for both in vitro and in vivo.
Lithography (TRD)		Graphene	0.5–120 µm	nm	Good sensitivity in urine samples
Screen-printed electrode (TRD)		Conducting polymer–Pd composite	0.1 to 200 μm	nm	In vitro sensing capabilities
3D printing (AJP)	Glucose	Polymer nanocomposite	0–10 mm	6.9 μm	Multimaterial printing, customizability, and rapid prototyping. High sensitivity.
3D printing (Inkjet printing)		PEDOT.PSS	0.25–0.9 mm	μm	Rapid, fully printed, and customizable biosensor. Noninvasive, good sensitivity in saliva, stability ≈ 1 month, and response time ≈ 1 min
Electrodeposition (TRD)		MnO2/MWCNTs	10μm–28 mm	μm	Low-potential, stable, and fast detection time
3D printing (SLM)	Ascorbic acid	Au electrode	0.1–1 mm	2.1 μm	Multimaterial printing and rapid prototyping.
Glassy carbon electrode (TRD)		Carbon nanoplatelets	0.1 µm–1.8 mm	1.09 μm	Sensing ability with soft drinks, orange juice, and urine.
Lithography (TRD)		Indium tin oxide (ITO) electrode	0.058 to 0.71 mm	8.4 μm	Response time ≈ 40 s, shelf life ≈ 1.5 months

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
