# Peer review of "The 3D Printing of Nanocomposites for Wearable Biosensors: Recent Advances, Challenges, and Prospects"

_bioengineering, 2023, doi:10.3390/bioengineering11010032_

Round 1

Reviewer 1 Report

Comments and Suggestions for Authors

The authors of this review present an overview of wearable biosensors, nanocomposites, the additive manufacturing (3D printing) processes, recent advances in additive manufacturing (3D printing) for wearable biosensors, and the general challenges that are now being encountered with 3D printed wearable biosensors. This manuscript could be considered for published once the following issues have been addressed.

1, Nanomaterials can be categorized into three types, so which nanomaterials are more suitable for compositing on 3D printed wearable biosensors? Or what are the advantages of compositing each kind of nanomaterials respectively.

2The fourth part cites a lot of applications of 3D Printing of Nanocomposites for Wearable Biosensors, and I think some comparisons with ordinary biosensors can be added as well as

3, In the section of challenge of this part of the many technical problems that hinder the full development of 3D Printing of Wearable Sensors, I think we can cite some 3D Printing of Wearable Sensors in the application of the specific problems encountered in the application of this part of the support and then give examples.

Comments on the Quality of English Language

This English could be improved at several parts. 

Author Response

Revision Report for Manuscript ID: 2757265 – Bioengineering
Title: 3D Printing of Nanocomposites for Wearable Biosensors: Recent Advances, 
Challenges, and Prospects.
We thank the reviewer for the constructive comments in improving the quality of this revised 
manuscript. Our report below outlines the detailed revisions undertaken in response to the 
comments raised.
Note:
· Reviewer comments and concerns are shown.
· Response on answers to questions is provided below in blue color.
· Corrections in the manuscript are highlighted in yellow color.
Review 1: 
Comments and Suggestions for Authors
The authors of this review present an overview of wearable biosensors, nanocomposites, the 
additive manufacturing (3D printing) processes, recent advances in additive manufacturing (3D 
printing) for wearable biosensors, and the general challenges that are now being encountered with 
3D printed wearable biosensors. This manuscript could be considered for published once the 
following issues have been addressed.
1, Nanomaterials can be categorized into three types, so which nanomaterials are more suitable 
for compositing on 3D-printed wearable biosensors? Or what are the advantages of compositing 
each kind of nanomaterials respectively?
The details about the nanocomposites for 3D printed wearable biosensors and their respective 
advantage were added as suggested by the reviewer in Ln202 – Ln218.
Nanocomposite materials demonstrate enhanced properties due to an exceptionally high 
aspect ratio. The integration of nano-particle reinforcements enables improved performance as 
compared to monolithic or micro composites. Nanomaterials which include nanorods, nanofibers, 
nanowires, and nanotubes are more promising for the different types of matrix nanocomposite 
materials for 3D printing of wearable biosensors [94]. These nanomaterials have a high surfaceto-volume ratio which makes them the ideal choice for sensing applications [87][90]. Based on the 
desired material properties respective nanocomposite materials (PMNC, CMNC, and MMNC) can 
be utilized for various applications such as biomedical, healthcare, solar cells, aerospace, 
supercapacitors, etc. MMNC materials improve the properties such as ductility and toughness of 
the materials. Therefore, MMNCs are more suitable for applications with high strength in 
shear/compression and extreme temperature abilities. The reinforcement of nanofibers in ceramic 
matrices enables the advancement of CMNC materials for application with superior failure 
properties and high toughness. In contrast, the nanofillers in the polymer matrix materials (PMNC) 
possess enhanced electrical conductivity and colloidal stability. PMMNC materials improve the 
biodegradability of the materials which is essential for biomedical and healthcare applications [86].
2、The fourth part cites a lot of applications of 3D Printing of Nanocomposites for Wearable 
Biosensors, and I think some comparisons with ordinary biosensors can be added as well as
The comparison of 3D printed biosensors with ordinary biosensors is provided as suggested by the 
reviewer. 
4.9 3D-Printed Wearable Biosensors vs Traditional Wearable Biosensors 
The comparison of the 3D printed biosensors with the traditional (TRD) biosensors is 
provided in Table 4. Based on the findings reported by various researchers in Table 4 it can be 
inferred that 3D-printed biosensors enable the detection of biomarkers with high sensitivity 
compared to traditional biosensors. (Ln 790 – Ln 807)
Table 4 provides a comparison of 3D-printed biomedical sensors with traditional sensors. 
Table 4. Comparison of 3D printed biosensors with traditional (TRD) biosensors © 2021 Wiley‐
VCH GmbH [43].
Methods Biomarker Sensor 
structures (3D & 
non-3D)
Detection 
range
LoD Sensing capabilities and 
remarks
3D printing 
(SLA)
Prostatespecific 
antigen
3D printed 
channels; 
immunoarray
0.5 pg 
mL−1 to 
5 ng mL−1
0.5 pg
mL−1
Customizability and rapid 
prototyping capability. 
Automated detection system 
and assay time ≈ 30 min. 
Accuracy comparable with 
ELISA and commercial 
devices such as Abbott 
Diagnostics (0.008 ng mL−1), 
Roche (0.002 ng mL−1), 
Beckman Coulter (0.008 ng 
mL−1), and Diagnostic 
Products Corporation (0.04 ng 
mL−1)
Commercial SPR 
biochip (TRD)
Self-assembled 
monolayered Au
1–1000 ng 
mL−1
18.1 
ng 
mL−1
Assay time ≈14 min. Sensing 
with buffer solution and 
human serum
Microfabrication 
(TRD)
Self-assembled 
monolayered Au
0-4 µg 
mL−1
0.2 µg 
mL−1
Single-use biosensor, sensing 
with serum samples and good 
sensitivity
3D Printing (AJP) Dopamine Micropillar array 
electrode
100 am–1 
mm
am Low LoD ≈ 500 attomoles, 
breaking the barrier described 
in literature[47] through 
multi-length-scale electrode 
structure. Rapid prototyping 
capability and waste 
minimization due to small 
microfluidic volume required 
for testing.
3D printing (2PP) 3D carbon 
electrode
0.5–
100 µm
nm High sensitivity to multiple 
neurochemicals, high 
reproducibility, capability for 
both in vitro and in vivo.
Lithography
(TRD)
Graphene 0.5–
120 µm
nm Good sensitivity in urine 
sample
Screen printed 
electrode (TRD) 
Conducting 
polymer-Pd 
composite
0.1 to 200 
μm 
nm In vitro sensing capabilities
3D printing (AJP) Glucose Polymer 
nanocomposite
0–10 mm 6.9 
μm
Multi-materials printing, 
customizability, and rapid 
prototyping. High sensitivity.
3D printing 
(Inkjet printing)
PEDOT.PSS 0.25–0.9 
mm
μm Rapid, fully printed, and 
customizable biosensor. Noninvasive, good sensitivity in 
saliva, stability ≈1 month, and 
response time ≈1 min
Electrodeposition 
(TRD)
MnO2/MWCNTs μm–28 
mm
μm Low-potential, stable, and fast 
detection time
3D printing 
(SLM)
Ascorbic 
acid
Au electrode 0.1–1 mm 2.1 
μm
Multi-materials printing and 
rapid prototyping.
Glassy carbon 
electrode (TRD)
Carbon 
nanoplatelets
0.1 µm–
1.8 mm
1.09 
μm
Sensing ability with soft 
drink, orange juice, and urine.
Lithography 
(TRD) 
Indium tin oxide 
(ITO) electrode
0.058 to 
0.71 mm 
8.4 
μm
Response time ≈40 s, shelf life 
≈1.5 month
3, In the section of challenge of this part of the many technical problems that hinder the full 
development of 3D Printing of Wearable Sensors, I think we can cite some 3D Printing of 
Wearable Sensors in the application of the specific problems encountered in the application of this 
part of the support and then give examples.
In the challenges section, the issues encountered by the 3D printing of wearable biosensors 
were cited appropriately relating to the issues encountered by the prior research literature as 
suggested by the reviewer. Figure 13 was added to provide the schematic of the 3D-printed 
wearable biosensors challenges encountered in the past and the potential setbacks. 
Figure 13. Schematic of 3D-printed wearable biosensors challenges.

Reviewer 2 Report

Comments and Suggestions for Authors

In this manuscript, the authors reviewed the development of 3D printing of nanocomposites for wearable biosensors, and also discussed the challenges and future prospects. This review seems to be useful in the related filed. However, the following problems should be addressed before further consideration of publication:

1. The keywords need reduction to be concise.

2. This review contains rich contents yet the structure needs revision for better demonstration. The authors are suggested to revise it into a more clear structure (Introduction, Various types of researches, Challenges and future prospects, Conclusions). A schematic figure can be created at the end to show the challenges and future prospects.

3. The Introduction is suggested to come straight to the point, and references should be enriched. The related advances and applications of various flexible sensors should also be added including: 10.1021/acsami.2c22727, 10.1002/EXP.20210033, 10.1021/acsami.2c10226.

4. Table 1 and 2 seem to be simple and incomplete, which need revision for better demonstration.

5. For a comprehensive review, the authors are suggested to make a brief summary of the used 3D printing methods including the mechanisms, process, and detailed applications of various applications.

6. In Section 4, the review is focused on various functions of wearable biosensors. The depth could be improved if the authors provided some insights of materials and micro/nano interactions responsible for biosensing.

7. The format of references should be check thoroughly considering the citing errors. 

Author Response

Revision Report for Manuscript ID: 2757265 – Bioengineering 
Title: 3D Printing of Nanocomposites for Wearable Biosensors: Recent Advances, 
Challenges, and Prospects.
We thank the reviewer for the constructive comments in improving the quality of this revised 
manuscript. Our report below outlines the detailed revisions undertaken in response to the 
comments raised.
Note:
· Reviewer comments and concerns are shown.
· Response on answers to questions is provided below in blue color.
· Corrections in the manuscript are highlighted in yellow color.
Review 2: 
Comments and Suggestions for Authors
In this manuscript, the authors reviewed the development of 3D printing of nanocomposites for 
wearable biosensors, and also discussed the challenges and future prospects. This review seems to 
be useful in the related field. However, the following problems should be addressed before further 
consideration of publication:
1. The keywords need reduction to be concise.
The keywords were reduced as follows: 
Keywords: 3D Printing; Biomedical; Health Monitoring; Nanocomposites; Wearable Biosensors
2. This review contains rich contents yet the structure needs revision for better demonstration. The 
authors are suggested to revise it into a more clear structure (Introduction, Various types of 
researches, Challenges, and future prospects, Conclusions). A schematic figure can be created at 
the end to show the challenges and future prospects.
The content in the manuscript is revised as per the reviewer's suggestion and the schematic figures 
were added to the manuscript in the appropriate section as shown below (Figures 13 & 14): 
Figure 13. Schematic of 3D-printed wearable biosensors challenges. (Ln 819)

Figure 14. Schematic representing the prospects of the wearable biosensors. (Ln 906)
3. The Introduction is suggested to come straight to the point, and references should be enriched. 
The related advances and applications of various flexible sensors should also be added including 
10.1021/acsami.2c22727, 10.1002/EXP.20210033, 10.1021/acsami.2c10226.
Minor revisions have been made to the introduction. References were enriched with the addition 
of new references in the appropriate section: Wearable biosensors such as wearable electronics, 
artificial skin, pressure sensors, glucose sensors, strain sensors, tattoo sensors, etc., have stimulated 
considerable importance in the healthcare arena for real-time health monitoring applications 
[112][113][114]. (Ln 334)
4. Table 1 and 2 seem to be simple and incomplete, which need revision for better demonstration.
The measuring range of the commercially available wearable biosensors is included in Table 1. 
Table 1. Some of the commercially available wearable biosensors [24]. (Ln 54)
Company Model Analyte Measuring range (mM)
Yellow Springs 
Instruments
23 A
13L
Glucose 
Lactate
1-45
0-15
Zentrum für 
Wissenschaftlichen
Geratebau, Berlin, 
Germany
Glucometer Glucose 
Uric acid
0.5-50
0.1-1.2
Abbott, USA FreeStyle 
InsuLinx Glucose, insulin 20–500 mg/dl
Lifestream cholesterol 
monitor Alcosan saliva 
alcohol dipstick
i-STAT
PCA
Glucose 
Urea nitrogen, Cl, K, 
Na+ , hematocrit blood 
gases
20-700 mg/dL
-
DEX blood glucose meter Bioscanner 
2000
Glucose, cholesterol, 
HDL, blood ketone, 
triglyceride
55-230mg/dl
-
Table 2 provides detailed examples of different types of composite materials in three different 
matrix nanocomposites with respective properties. 
Table 2. Different types of nanocomposites [90]. (Ln 166)
Class Examples Properties
Polymer
Thermoplastics/layered silicates/thermoset 
polymers, polymer/CNT, polyester/TiO2, polymer/ 
layered double hydroxides.
Enhanced electrical conductivity and colloidal 
stability, Biodegradability
Ceramic Al2O3/CNT Polymer, Al2O3/TiO2, SiO2 /Ni, 
Al2O3/SiC, Al2O3/SiO2
High Toughness and 
Superior Failure Properties
Metal Fe-Cr/Al2O3, Al/CNT, Mg/CNT, Co/Cr, Fe/MgO, 
Ni/Al2O3 
Strong Ductility and 
Excellent shear/compression practices
5. For a comprehensive review, the authors are suggested to make a brief summary of the 
used 3D printing methods including the mechanisms, process, and detailed applications of 
various applications.
A summary of different 3D printing processes is provided in Section 3: Additive Manufacturing 
(3D Printing) Processes as suggested by the reviewer (Ln 236 -Ln 262): 
Fused deposition modeling techniques have applications in hearing aids, surgical 
instruments or tools, dental implants, and pharmaceutical applications. Stereolithography 
techniques have applications in prosthetics, implantable devices, orthodontics, surgical planning, 
drug delivery, and dentistry. The selective laser sintering process has an application in tissue 
engineering. orthopedics, and surgical instruments or tools. Binder jetting technique applications 
can be found in surgical instruments or tools, medical models, and drug implant prosthetics. The 
electron beam melting process has applications in hip and knee replacements, cranial implants, 
dental implants, and surgical instruments or tools. Digital light processing has applications in tissue 
engineering, surgical planning, and medical imaging. The most popularly used 3D printing 
techniques for wearable biosensors are fused deposition modeling, selective laser sintering, inkjet 3D printing, stereolithography, two-photon polymerization, and laminated object 
manufacturing [101]. A detailed overview of different types of 3d printing processes with technical 
details is presented in Table 3.
6. In Section 4, the review is focused on various functions of wearable biosensors. The depth 
could be improved if the authors provided some insights of materials and micro/nano 
interactions responsible for biosensing.
The insights of materials and micro/nano interactions responsible for biosensing were
provided in Section 2: Nanocomposites (Ln218-Ln234) as suggested by the reviewer. This 
overview of the material interaction applies to all wearable devices discussed in Section 4 
Nanomaterials are capable of both increasing the detection of moieties and accelerating 
transduction signals. Moreover, the slightest change in the response can be captured effectively. 
The major objective of incorporating nanoparticle materials is to elevate biosensor accuracy and 
efficiency by augmenting sensor responses, considerably lowering detection limits (LOD), and 
reducing processing times. Due to the unique capabilities of nanomaterials such as their distinctive 
size and shape-dependent attributes, particularly in energy band-gap modulation, these materials 
provide rapid and precise analyte detection. For example, traditional glucose detection techniques 
can be enhanced significantly by functionalizing enzymes on graphene sheets, or carbon nanotubes 
which in turn increases the detection sensitivity and reduces LOD. Furthermore, the incorporation 
of nanomaterial-based alternatives such as polymer/metal/ceramic nanoparticles, quantum dots, 
nanowires, and nanorods could result in improved performance of wearable biosensor devices 
[95]. These remarkable abilities of nanomaterial interactions with the analyte enable effective 
biosensing with an emphasis on biomedicine and healthcare domains. The utilization of 
nanomaterials in biosensing technology represents a cutting-edge advancement in detecting 
capabilities for the wearable biosensor [92]. 
7. The format of references should be checked thoroughly considering the citing errors. 
The format of the references was reviewed thoroughly and corrected appropriately in the whole 
manuscript. 

Reviewer 3 Report

Comments and Suggestions for Authors

Review report – 2757265 – Bioengineering

A brief summary

Goal of this study was to comprehensively review the progress in 3D-printed wearable biosensors. Among other goals the review also explores the incorporation of nano-composites in 3D printing for biosensors.

Broad comments

Introducing overview was supported by methodology used to fulfil aim of the review. Design was appropriate, followed and supported by relevant and precise conclusions. Limitations were sufficiently explained. 

Significant developments of 3D printing processes in the past decade for manufacturing novel next-generation wearable biosensor constructs using nano-composite materials were outlined in this very high-standard review. However, since other biosensing-based solutions (related to products from e.g. Neuralink, Robotic Neurorehabilitation, Spine Bridge, Synestesia-based vision, sEMG, methylation and likewise) were not included, it doesn’t need any major revision, but small intervention into the title: “3D Printing of Nanocomposites for SELECTED or STANDARDIZED Wearable Biosensors: Recent Advances, Challenges, and Prospects.” It would greatly affirm authors’ in-depth approach. Section ‘Challenges’ envelopes this affirmation concisely. 

Specific comments.

·       Minor spell check, and style corrections needed. 

·       Ln 93 ‘low cost, minimal cost’ into low/minimum cost

·       Ln 267/8 ..’printing processses, …printing speed,… printing’ – preformulate!

·       Ln33/2 Figure 5. -within b) (“DLP printed conductive hydrogel © 2011 Elsevier Ltd. “) sEMG was depicted, however insufficiently covered 

·       Within Figure 8, within a) – b,c,d were correctlly presented but ‘a’ is missing

·       Ln 777 – with regard to power consumption-reformulate please. There is a whole spectrum of inertial-magneto-electrochemical solutions that go beyond movement (applied in old watches e.g.), e.g. skin surface micro-voltage harvesting etc..

Comments on the Quality of English Language

few inconsistencies within text are in ‘Specific comments’ - follow the pattern within whole text. 

Author Response

Revision Report for Manuscript ID: 2757265 – Bioengineering 
Title: 3D Printing of Nanocomposites for Wearable Biosensors: Recent Advances, 
Challenges, and Prospects.
We thank the reviewer for the constructive comments in improving the quality of this revised 
manuscript. Our report below outlines the detailed revisions undertaken in response to the 
comments raised.
Note:
· Reviewer comments and concerns are shown.
· Response on answers to questions is provided below in blue color.
· Corrections in the manuscript are highlighted in yellow color.
Review 3: 
Comments and Suggestions for Authors
Review report – 2757265 – Bioengineering
A brief summary
Goal of this study was to comprehensively review the progress in 3D-printed wearable biosensors. 
Among other goals the review also explores the incorporation of nanocomposites in 3D printing 
for biosensors.
Broad comments
Introducing overview was supported by the methodology used to fulfil aim of the review. Design 
was appropriate, followed, and supported by relevant and precise conclusions. Limitations were 
sufficiently explained. 
Significant developments of 3D printing processes in the past decade for manufacturing novel 
next-generation wearable biosensor constructs using nano-composite materials were outlined in 
this very high-standard review. However, since other biosensing-based solutions (related to 
products from e.g. Neuralink, Robotic Neurorehabilitation, Spine Bridge, Synestesia-based vision, 
sEMG, methylation and likewise) were not included, it doesn’t need any major revision, but small 
intervention into the title: “3D Printing of Nanocomposites for SELECTED or 
STANDARDIZED Wearable Biosensors: Recent Advances, Challenges, and Prospects.” It would 
greatly affirm authors’ in-depth approach. Section ‘Challenges’ envelopes this affirmation 
concisely. 
In this review we have reported the recent advancements in 3D-printed wearable biosensors, 
highlighting their contributions and applications to healthcare and personalized monitoring. In 
section 4 of our paper, we present the research progress of 3D-printed wearable biosensors which 
include electrocardiogram (ECG), electroencephalogram (EEG), blood pressure, glucose, oxygen 
saturation (SpO2), sweat, textile, and respiratory biosensors are presented. The above-mentioned 
wearable biosensors represent in-field solutions where 3D-printed nanocomposites are 
implemented. The other biosensing solutions based on commercial products mentioned by the 
reviewer may or may not necessarily be amenable to 3D-printed nanocomposites and thus beyond 
the scope of the current focus area of the paper. The scope of this review article is limited to the 
above-mentioned wearable biosensors and thus, we would like to retain the current title of the 
paper. 
Specific comments.
· Minor spell check, and style corrections needed. 
The whole manuscript spell and style corrections were addressed as suggested by the reviewer.
· Ln 93 ‘low cost, minimal cost’ into low/minimum cost
Biosensors are suitable for wearable applications because of their low/minimal cost, high 
specificity, and nominal power requirements (Currently in Ln 82). 
· Ln 267/8 ..’printing processses, …printing speed,… printing’ – preformulate!
Many of the above-mentioned 3D printing technologies are being established gradually because 
of the significant enhancements in manufacturing processes, feature size, printing speed, 
material, and resolution with respect to layer fusion nature (Currently in Ln 321-3223).
· Ln33/2 Figure 5. -within b) (“DLP printed conductive hydrogel © 2011 Elsevier Ltd. “) 
sEMG was depicted, however insufficiently covered 
Figure 5 (b) image was replaced with the original image acquired from the respective journal 
article. [(b) DLP printed conductive hydrogel © 2011 Elsevier Ltd. [126]]
· Within Figure 8, within a) – b,c,d were correctlly presented but ‘a’ is missing
All the missing details were added to Figure 8. (a) Smartphone-based 3D-printed 
electrochemiluminescence enzyme biosensor © 2023 Calabria et.al. Published by Elsevier B.V 
[167]
· Ln 777 – with regard to power consumption-reformulate please. There is a whole spectrum 
of inertial-magneto-electrochemical solutions that go beyond movement (applied in old watches 
e.g.), e.g. skin surface micro-voltage harvesting etc..
We agree with the reviewer that there are several methods to generate electric power beyond 
battery sources. We have incorporated these within the Power Consumption sub-section of 5.0 
Challenges with details below: (Ln 859 -Ln 872)
Power consumption: This issue can be addressed by utilizing energy storage components and selfpowered devices to ensure that the 3D-printed wearable biosensors are supplied with the necessary 
power to monitor the human elements continuously without any interruptions. In addition, 
miniaturization of the power circuitry and harnessing power from patient movements can extend 
the life of batteries for wearable devices. Energy can be harvested from the human or the ambient 
environment to prolong the working life of wearable devices [220]. Inertial systems have been 
used to generate electricity from extremity movements such as hands, legs, and other body parts 
[221]. The hu-man-centric energy harvesting methods can range from both biochemical and 
biomechanical methods. Environment-centric energy harvesting can include infrared radiation, 
radio-frequency signals, and solar energy to name a few [222]. Further, a hybrid approach can be 
implemented with a combination of the above-mentioned technologies to generate sustained levels 
of electric power. Microscale 3D printing can aid in the fabrication of microelectromechanical 
(MEMs) devices which have been traditionally done by lithography-based methods [223]. 3D 
printed microscale can enable hierarchical multilayered structures as well as conformal [224] 
conductive traces for compact devices which can be contained within the wearable device 
footprint.
Comments on the Quality of English Language
few inconsistencies within text are in ‘Specific comments’ - follow the pattern within whole text. 
The overall quality of the English language was modified to enhance the manuscript writing as 
suggested by the reviewer. 

Round 2

Reviewer 2 Report

Comments and Suggestions for Authors

The paper is well revised. I think it can be accepted now.